# Comprehensive Analysis of Compressible Perceptual Encryption Methods—Compression and Encryption Perspectives

**DOI:** 10.3390/s23084057

**Published:** 2023-04-17

**Authors:** Ijaz Ahmad, Wooyeol Choi, Seokjoo Shin

**Affiliations:** Department of Computer Engineering, Chosun University, Gwangju 61452, Republic of Korea

**Keywords:** perceptual encryption, JPEG compression, encryption-then-compression schemes

## Abstract

Perceptual encryption (PE) hides the identifiable information of an image in such a way that its intrinsic characteristics remain intact. This recognizable perceptual quality can be used to enable computation in the encryption domain. A class of PE algorithms based on block-level processing has recently gained popularity for their ability to generate JPEG-compressible cipher images. A tradeoff in these methods, however, is between the security efficiency and compression savings due to the chosen block size. Several methods (such as the processing of each color component independently, image representation, and sub-block-level processing) have been proposed to effectively manage this tradeoff. The current study adapts these assorted practices into a uniform framework to provide a fair comparison of their results. Specifically, their compression quality is investigated under various design parameters, such as the choice of colorspace, image representation, chroma subsampling, quantization tables, and block size. Our analyses have shown that at best the PE methods introduce a decrease of 6% and 3% in the JPEG compression performance with and without chroma subsampling, respectively. Additionally, their encryption quality is quantified in terms of several statistical analyses. The simulation results show that block-based PE methods exhibit several favorable properties for the encryption-then-compression schemes. Nonetheless, to avoid any pitfalls, their principal design should be carefully considered in the context of the applications for which we outlined possible future research directions.

## 1. Introduction

Image data transmission has the dual requirements of compression and encryption, like any other type of data. Compression is a process that reduces the data size by exploiting redundancies (such as spatial and psycho-visual redundancies) present in an image, whereas encryption makes an image unintelligible by adding randomness to it. Thereby, both are related but inverse processes, and the order in which they are coupled together results in a tradeoff between compression and security efficiencies. The conventional order is to perform compression prior to encryption, compression-then-encryption (CtE) methods, as completing encryption before compression will destroy the image correlation. In this regard, traditional number theory and chaos theory-based encryption algorithms are proven to be secure for the protection of multimedia content [1,2]. The CtE methods perform pixel scrambling or stream encryption and are mainly applicable for the encryption of raw images. However, they are not adequate to encrypt compressed images while preserving the compression savings, image format, and providing the necessary level of security. For example, when encrypting a JPEG image, this operation can disturb JPEG format identifiers, which may lead to certain issues such as format incompatibility and an increment in the file size. Any changes to the JPEG markers may render them uninterpretable and re-encoding the cipher text as a JPEG image will increment the image size. Image format compliancy is necessary for cloud-based photo storage services (CPSS), social networking services (SNS), reversible data-hiding applications, and image processing in the encryption domain (such as image retrieval, privacy-preserving machine learning (PPML), etc.).

Another way to couple compression and encryption is the joint operation of performing encryption within compression, encryption-in-compression (EiC) methods. However, there are certain limitations based on the position of the encryption algorithm in the compression. For example, encryption can be achieved by using multiple new orthogonal transforms during the transformation stage, as proposed in [3,4]. The schemes deliver a better tradeoff between compression and encryption. However, in [3], the security strength is limited, as only the transformation is modified and the lack of diffusion property makes them vulnerable to differential attacks [5]. On the other hand, the scheme proposed in [4] has better coding efficiency than [3]; however, the block-level processing limits the decorrelation abilities and makes it vulnerable to statistical attacks [5]. An alternative way is to perform encryption in the quantization step either by scrambling quantized DC and AC coefficients [6] or by changing the magnitudes of entries in the quantization table [7,8]. The main advantage of these methods is format compliancy. The scheme proposed in [6] preserves almost the same compression savings; however, it is vulnerable to non-zero-counting attack. The methods proposed in [7,8] are not secure enough, and the compression ratio also suffers. In [9], efficient security and format compliancy is achieved by encrypting only DC coefficients and the first 14 AC coefficients in the zigzag scan. However, the compression savings are heavily compromised. Alternatively, encryption can be achieved in the intermediate encoding step of the JPEG compression. For example, Ref. [10] proposed that instead of using the standard zigzag scan, the DCT coefficients can be scanned by using different patterns. Such methods provide better security with good diffusion and confusion properties. However, their main limitations are a high computation cost and an increment in file size. To achieve a better compression and encryption tradeoff, Ref. [11] proposed the encryption of selected coefficients specified by a range and the shuffling of the block identifier positions. However, this leads to a format incompatibility issue. Finally, in the entropy encoding stage, one way to achieve encryption is to use multiple Huffman tables [12]. Because the probability distribution of the image is left unaltered by the encryption process, the compression savings are preserved. However, during decoding of the cipher image, all the Huffman tables should be made available to the decoder, which results in a format compliancy problem. In addition, such schemes are vulnerable to known- and chosen-plaintext attacks [13]. An alternative method is proposed in [8] to encrypt the output bitstream of the Huffman encoder while keeping the Huffman codes unmodified. The method preserves both the files size and image format. However, leaving the Huffman codes in plain makes the method vulnerable to image contour reconstruction attack. In [14], the authors proposed to assign each Huffman codeword to another codeword with the same code length to carry out encryption. The method is compression friendly; however, adopting their mapping strategy leads to a format compliancy problem. Refs. [15,16] proposed selective encryption of the DCT coefficients; however, it requires knowledge of the important coefficients beforehand. A hybrid compression encryption is proposed in [17] based on chaos theory, but it does not consider JPEG compression standard.

JPEG image encryption has requirements of format compliance, reasonable security and small file size increment. The CtE and EiC methods are unable to meet these requirements, as discussed earlier. An alternative approach is to perform encryption before compression, encryption-then-compression (EtC) methods. The main challenge of reversing the CtE order is the preservation of compression savings, as the encryption process disturbs the image correlation [18]. However, the methods proposed in [19,20,21,22,23,24,25,26,27,28,29] have shown that compression of the encrypted images can be achieved with a slight degradation or even with the same compression savings. The methods provide a necessary level of security, but they are not JPEG compatible. In recent years, a new image encryption algorithm has been proposed to hide only the perceptual details of an image while retaining its intrinsic properties necessary for compression. The methods belong to the EtC class, and in this paper, we referred to them as compressible perceptual encryption methods—CPE for short. The encryption algorithm is block based and performs four steps: block permutation, block rotation, block inversion, and pixel level negative–positive transformation. Nonetheless, these schemes are robust against various types of attacks including brute-force attack and cipher-text only attack. The encryption algorithm is computationally inexpensive and is JPEG compatible, thereby suitable for CPS and SNS services, image retrieval systems, and PPML applications and medical image services.

Several studies have improved the encryption efficiency of the CPE schemes. For example, Ref. [30] proposed a CPE scheme with an additional step to permute the blocks in the color channels for improved encryption efficiency. However, this scheme has a limitation on the keyspace size resulting from the choice of block size. The smallest block size that can be used is 16 × 16 to avoid distortion in the recovered image when JPEG chroma subsampling is being used. In [31], the authors proposed to process each color component independently for a larger keyspace size. However, the methods are only compatible with the JPEG lossless compression standard. To deal with these issues, Refs. [32,33] proposed to represent the input image as a grayscale image by combining the color channels along the horizontal or vertical direction. Such representation allows the use of a smaller block size of 8 × 8, thus improving the encryption efficiency. The methods proposed in [30,31,32,33] have color image input as a prerequisite for better encryption efficiency. In [34], the authors proposed sub-block processing for the efficient encryption of grayscale images. However, in the CPE schemes, there is a tradeoff between encryption and compression efficiency because of the block size. For efficient encryption, a larger number of blocks is desirable to expand the keyspace [35].

In this paper, we present a comprehensive analysis of the JPEG-compatible CPE schemes in terms of their encryption and compression efficiencies. The existing surveys in the literature are either focused on the image encryption techniques that are applicable for raw image protection [1,2,36,37] or nonstandard image compression formats [38,39]. To the best of our knowledge, Refs. [5,40,41] are the most related surveys to the current study that deal with the JPEG-compatible perceptual encryption schemes. In [40,41], the authors studied CPE and noncompressible perceptual encryption methods mainly from a PPML application point of view. On the other hand, the authors in [5] focused on joint compression and encryption algorithms in general and covered only two CPE schemes. Different from the existing surveys, the main contributions of the current survey can be summarized as follows: (1) An evaluation of compression performance under various conditions, such as input image representation, colorspace conversion, quantization table choice, and compression with and without chroma subsampling, is performed in this study. (2) In the literature, the compression savings of the methods were subjectively analyzed using only peak signal-to-noise ratio (PSNR)-based rate distortion (RD) curves. On the contrary, the current study uses better image quality metrics, such as multiscale structural similarity index measure (MS–SSIM), and objectively compares the RD curves using Bjøntegaard delta (BD) metrics. (3) In the literature, security efficiency of the CPE schemes was analyzed by showing robustness against a jigsaw puzzle solver (JPS) attack only. In contrast, the current study compares the CPE methods using differential attack analysis, histogram variance analysis, entropy analysis, and correlation coefficient analysis along with the keyspace size analysis and robustness against the JPS attack.

The rest of the paper is summarized as follows: Section 2 presents the related work on CPE schemes along with their applications. Section 3 provides preliminary details including the JPEG image standard. Section 4 gives an overview of the CPE methods. In Section 5, several CPE schemes were implemented under different conditions and compared for their compression and encryption performance efficiencies. Section 6 discusses the CPE scheme advantages with respect to the application requirements and gives future research directions. Section 7 concludes the paper.

## 2. Related Work

Figure 1 shows a taxonomy of image encryption methods, which classifies them into full encryption and partial encryption methods. The full encryption methods hide all the information of an image and comprise the traditional number theory- and chaos theory-based algorithms. The partial encryption methods hide only selected information in an image, for example, the selective encryption algorithms only protect the region of interest in an image, whereas the perceptual encryption algorithms only hide the human perceivable and identifiable information in an image. The perceptual encryption algorithms can be further classified as incompressible methods, which perform pixel level scrambling, and compressible methods, which process image blocks. In Figure 1, from left to right, the encryption algorithms computational complexity decreases and security is traded to enable other multimedia applications such as format compliant storage and even processing the encryption domain. The main focus of the present study are the perceptual encryption methods, specifically, the block-based compressible methods.

In general, the encryption algorithm of a CPE scheme is block-based and consists of four steps: block permutation, block rotation, block inversion, and negative and positive transformation. There is an optional color-channel shuffling step that is used when the input is a color image. The existing CPE methods can be classified based on their input image representation, such as Color CPE, Extended CPE, inter and intra block processing-based CPE (IIB–CPE) and pseudo-grayscale-based CPE (PGS–CPE) methods. In the Color CPE, Extended CPE, and IIB–CPE methods, an input color image is represented by its three color components, whereas in PGS–CPE methods, the color components of an input color image are concatenated along the horizontal or vertical direction to form a pseudo-grayscale image. An alternative classification of CPE methods is based on their mode of processing, for example, methods that transform an entire block include the Color CPE, Extended CPE, and PGS–CPE methods, and methods that incorporate sub-block processing include the IIB–CPE methods. This CPE classification is beneficial when the input is a grayscale image. The following subsections present the related work on each category along with their applications.

### 2.1. Color CPE Methods

Watanabe et al. proposed a Color CPE method that performs a color-channel shuffling step for better security, and their method is compatible with the JPEG 2000 standard [42] and the motion JPEG 2000 standard [43]. The applications of their method have been further extended by Kurihara et al. to the JPEG standard [30], the motion JPEG standard [44], the JPEG XR standard [45], and lossless image compression standards [46]. The Color CPE methods process image blocks with the same key in each color channel. The methods use a block size of 16 × 16 in the encryption algorithm to take advantage of the JPEG chroma subsampling step for better compression savings without any adverse effects. These methods preserve the JPEG file format and almost the same compression savings. However, the use of the common key to encrypt each channel leaves the color distribution unaltered, and the larger block size results in a smaller keyspace. This information makes the Color CPE schemes vulnerable to JPS attack [31].

### 2.2. Extended CPE Methods

To alter the color distribution in the Color CPE methods efficiently, Imaizumi et al. [31,47] proposed to process each color component individually in the permutation, rotation, inversion, and negative–positive transformation steps. This independent processing expands the keyspace size and modifies the color distribution significantly; however, this results in JPEG format compatibility issues. The main reason is that the JPEG standard requires colorspace conversion prior to compression and the Extended CPE methods are not suitable for this conversion function.

### 2.3. PGS–CPE Methods

In order to deal with the issue of Extended CPE methods, Chuman et al. proposed in [33] to perform the JPEG colorspace conversion prior to the encryption process. In addition, they proposed to concatenate the color components along the horizontal or vertical direction to form a pseudo-grayscale image. This grayscale representation can benefit from the smallest allowable block size, i.e., the JPEG performs a grayscale image compression on an 8 × 8 block size. This use of a small block size results in a larger keyspace size than the Color CPE and Extended CPE schemes. However, the PGS–CPE method proposed in [33] is not suitable for the JPEG chroma subsampling function. To deal with this issue, Sirichotedumrong et al. proposed in [32,48] to perform both the JPEG colorspace conversion and chroma subsampling functions prior to the encryption. The idea is to downsample the color components after the colorspace conversion and concatenate them with the luminance component. In addition, they proposed custom quantization tables in [48] that can be used in the JPEG standard for better compression performance.

### 2.4. IIB–CPE Methods

The Extended CPE and PGS–CPE methods have improved the security efficiency of the Color CPE methods, as the color distribution is scrambled significantly and the keyspace is expanded (especially in PGS–CPE methods). However, these schemes have a prerequisite of a color image as an input, for example, to achieve a large number of blocks, the individual color component processing (Extended CPE methods) and the pseudo-grayscale image representation (PGS–CPE methods) are only possible when the input is a color image. This advantage of these methods diminishes when the input image is a grayscale image with only one channel [49]. To overcome this limitation, Ahmad et al. proposed in [34,49,50] an inside-out transformation function that performs the rotation and inversion step on a sub-block level. Compared to the CPE methods that transform an entire block, these methods have a larger keyspace size for grayscale image processing. However, the methods are not suitable when the JPEG algorithm is implemented with the chroma subsampling function for color image compression.

Overall, in the CPE schemes—block-based perceptual encryption methods—there is an efficiency tradeoff between encryption and compression efficiencies because of the choice of block size. Specifically, a block size of no smaller than 16 × 16 and 8 × 8 should be used when considering the compression efficiency of the JPEG standard for color and grayscale images, respectively.

### 2.5. CPE Scheme Applications

The CPE schemes are suitable for privacy-preserving applications such as privacy-preserving photo sharing and storage services, privacy-preserving image retrieval systems, and PPML applications. In addition, the CPE schemes can also be used for reversible data-hiding applications.

Privacy-preserving photo sharing and storage applications: A privacy-preserving image trading system was proposed in [51] that uses the Color CPE algorithm of [30] for image copyright protection. In [52,53], the authors extended the applications of the Color CPE scheme in [30] to privacy-preserving photo sharing over third-party provided SNS. The main challenge in such applications are the artifacts resulting from the recompression of images by the SNS provides. The authors in [53] determined some parameters that can be used in order to resist such manipulations. Similarly, photo-sharing schemes based on an extended algorithm of the Color CPE and of the PGS–CPE were proposed in [54,55] and [56], respectively. The main advantage of the schemes was the identification of images re-encrypted with different keys. In [34,50], the authors proposed privacy-preserving photo storage for medical image applications based on an IIB–CPE scheme.

Privacy-preserving image retrieval applications: The CPE scheme’s cipher images preserve the image local contents on a block level; this information can be exploited for image retrieval applications without revealing the visual information of the image, as demonstrated in [57,58,59,60]. To achieve security, they used a Color CPE scheme with the JPEG and JPEG–LS standards.

Privacy-preserving computations applications: In [61,62], the authors identified a novel property of the CPE schemes that allows the computation of machine learning algorithms, such as support vector machines (SVM), in the encryption domain. They have shown that under different transformation functions of the CPE schemes, both the Euclidean distance and inner product of two vectors are preserved. In their experiments, they used a Color CPE algorithm without the color shuffling step for face recognition in a grayscale image dataset. Their analysis showed that the CPE schemes have no effect on the performance of the SVM algorithm. In similar work presented in [63], the authors used an Extended CPE method for face recognition in a color image dataset. Besides face recognition tasks, CPE-based privacy-preserving image classification has been performed in [49,64,65,66]. Specifically, in [64], the authors implemented an isotropic network such as vision transformers with the Color CPE scheme for natural image classification. In [65], the authors implemented four different extensions of IIB–CPE and analyzed their effect on a CNN model’s accuracy. The same authors implemented a CNN-based model with a IIB–CPE scheme for natural image classification in [49] and for COVID-19 diagnosis in chest X-ray images in [66].

Reversible data-hiding applications: In [67,68,69,70,71], the authors have proposed reversible data-hiding schemes using CPE cipher images. Retrieving the original image reversibility is an essential requirement of any data-hiding algorithm [69]. Therefore, to meet this requirement, the lossless JPEG standard should be used. Though both Color CPE and Extended CPE schemes are suitable for these applications, the data-hiding methods proposed in [67,68,69,70,71] are based on the Extended CPE methods to benefit from the larger keyspace size for efficient encryption.

## 3. Preliminaries

### 3.1. Notation Convention

Throughout this paper, scalars are denoted by italic letters x, row vectors by boldface letters $x=\left[x_{1},\cdots ,x_{N}\right]$, and matrices by capital boldface letters X, where $x_{i,j}$ represents the entry of X at row i, column j. The transpose of a matrix/vector is denoted by ${\left[\cdot \right]}^{\prime }$. Matrices are sometimes expressed in the compact form $X=\left[x_{1};x_{2};\cdots ;x_{M}\right]$, where $x_{i}=\left[x_{i,1},\cdots ,x_{i,N}\right]$ is the ith row. Sets are denoted using script letters S.

### 3.2. Image Block Partition

For a convenient representation of image partitioning, the number of rows and columns of an image $I_{H,W}$ can be represented as a product of two integers such as H=L×N rows and WHH=M×N columns. The image, therefore, can be divided into L×M blocks each with N×N pixels. The blocks can be represented in this image as $B_{i,j}$ with (i=0,1,⋯,L−1, j=0,1,⋯,M−1) where the (i,j) pair corresponds to (x,y) entry of the original image with some offset. For sub-block partitioning of a block $B_{N,N}$, its number of rows and columns can be represented in the same way as N=SL×SN. Consequently, this block will have SL×SL sub-blocks, each with SN×SN elements and denoted as $SB_{s,t}\;\left(s,t=0,1,\cdots ,SL-1\right)$, where the (s,t) pair corresponds to (i,j) entry of the block with some offset.

### 3.3. The JPEG Image Standard

The JPEG compression standard is one of the most widely used image formats. A block diagram of the JPEG algorithm is illustrated in Figure 2. The JPEG compression and decompression procedures can be described in the following steps.

Step 1. Colorspace Conversion

In the first step, the luminance component of an input image is separated from its color component, which is necessary to achieve more compression savings. The human visual system (HVS) is less sensitive to color than the image luminosity; therefore, the JPEG algorithm represents the color component in a smaller resolution; thus, it achieves more savings [72]. This process is called color or chroma subsampling. The ratio for chroma-subsampling depends on the application requirements; however, the most commonly used ratios are 4:2:2 (half of the color) and 4:2:0 (quarter of the color). The image luminance component (***Y***) can be separated from the image color components (***C_b_*** and ***C_r_***) by a colorspace conversion function defined as
(1)$$\{\begin{array}{c} Y=0.3\times R+\left(0.59\times G\right)+\left(0.11\times B\right)\\ C_{b}=128-\left(0.17\times R\right)-\left(0.33\times G\right)+\left(0.5\times B\right)\\ C_{r}=128+\left(0.5\times R\right)-\left(0.42\times G\right)-\left(0.08\times B\right) \end{array},$$
where R is the red, G is the green, and B is the blue color channel of the image. The Equation (1) converts an image from the RGB colorspace to the YCbCr colorspace. During decoding, an inverse operation is performed that converts the YCbCr image back to an RGB image, and this operation is defined as
(2)$$\{\begin{array}{c} R=Y+1.40\times \left(C_{r}-128\right)\\ G=Y-0.34\times \left(C_{b}-128\right)-0.71\times \left(C_{r}-128\right)\\ B=Y+1.77\times \left(C_{b}-128\right) \end{array}.$$
Note that when chroma–subsampling is performed during compression, then it is necessary to up sample the color components before the YCbCr to RGB conversion function during decompression to recover the full resolution image.

Step 2. Discrete Cosine Transformation (DCT)

The YCbCr image is divided into non-overlapping blocks, and each block is then transformed using the DCT function [73]. The goal here is to represent a large amount of information from a few data samples by exploiting the correlations among the adjacent pixels. In natural images, the pixels are usually high correlated up to 8 pixels neighbors in either direction [17]. Therefore, in the JPEG standard, a block size of 8 × 8 is used. The forward DCT function for the image block B can be defined as [72]
(3)$$F_{u,v}=\frac{1}{4}\alpha \left(u\right)\alpha \left(v\right)\left[\overset{7}{\underset{i=0}{\sum }}\overset{7}{\underset{j=0}{\sum }}B_{i,j}\times \cos\frac{\left(2x+1\right)u\pi }{16}\cos\frac{\left(2y+1\right)v\pi }{16}\right]\;\mathrm{where}\;\alpha \left(u\right),\alpha \left(v\right)=\{\begin{array}{cc} \frac{1}{\sqrt{2}} & u,v=0\\ 1 & otherwise \end{array}.$$
The result of the DCT function for an 8 × 8 image block is a 64 coefficient matrix that contains the 2D spatial frequencies. The element (0,0) in the matrix is called “DC coefficient” and has zero frequency in both directions. The remaining 63 elements are called the “AC coefficients”, for which the frequencies increase from left top corner to the right bottom corner in the matrix [72]. The inverse function of Equation (3) during decompression can be defined as
(4)$${\overset{ˇ}{B}}_{i,j}=\frac{1}{4}\left[\overset{7}{\underset{u=0}{\sum }}\overset{7}{\underset{v=0}{\sum }}\alpha \left(u\right)\alpha \left(v\right)F_{u,v}\times cos\frac{\left(2i+1\right)u\pi }{16}cos\frac{\left(2j+1\right)v\pi }{16}\right].$$

Step 3. Quantization

As a result of the DCT function, most of the image contents are preserved in a few coefficients (low frequency), mostly in the top left corner of each block. The rest of the DCT coefficients corresponding to the higher frequencies are visually insignificant psycho-visual redundancies and can be discarded. Therefore, the next step in the JPEG compression is quantization, which divides each DCT coefficient by its corresponding element given in a 64-element quantization table (***QT***). The quantization step is controlled by a scalar value known as the JPEG quality factor (*qf*). The range is [0, 100], where 0 represents the lowest and 100 represents the highest quality image. The quantization function of the JPEG compression can be defined as
(5)$${\hat{F}}_{u,v}=round\left(\frac{F_{u,v}}{QT_{u,v}}\right).$$
The JPEG standard includes two quantization tables, one for each of the luminance and chrominance components given in Table 1 and Table 2. The standard tables are specified for *qf* = 50, from which other tables can be calculated. In addition, these tables can also be user-defined input to the encoder. Examples of custom quantization tables proposed in [48] that are used for the PGS–CPE cipher image compression are given in Table 3 and Table 4. During decoding, the inverse function of Equation (5) simply performs a multiplication operation to estimate the closest representation of the original DCT values as
(6)$${\overset{ˇ}{F}}_{u,v}={\hat{F}}_{u,v}\times QT_{u,v}.$$

Step 4. Intermediate Encoding

In this step, the quantized DCT coefficients are represented in such a way that more compression savings can be achieved in the final step. First, the coefficients ${\overset{ˇ}{F}}_{u,v}$ of each block are scanned in a zigzag order onto a vector called the Minimum Code Unit (MCU). As a result, zeros corresponding to the higher frequencies end up together and can be encoded in an efficient way, i.e., an End of Block (EOB) symbol is added to the MCU after the last non-zero coefficient. The DC and AC coefficients have different properties; thus, the DC coefficient is treated differently from the rest of the 63 AC coefficients. The DC coefficients of adjacent blocks have a higher correlation; therefore, the coefficients are differentially pulse code modulated (DPCM) with each other. A prediction error between the adjacent DC coefficients is encoded as the amplitude value $A_{{\overset{ˇ}{F}}_{u,v}},\;\left(u,v=0\right)$ of the coefficient in ones complement form. The size category of the prediction error is included in the head $H_{{\overset{ˇ}{F}}_{u,v}},\left(u,v=0\right)$ of the coefficient. The quantized AC coefficients are run-length encoded (RLC) such that the consecutive zero coefficients are compressed. The non-zero coefficients are encoded as [(run length, size), amplitude], where run length is the number of zeros between two consecutive non-zero AC coefficients and size is the number of bits required to represent the amplitude. The run length together with size are encoded as head $H_{{\overset{ˇ}{F}}_{u,v}},\left(u\neq 0,v\neq 0\right)$ of the coefficient. The value of the coefficient is encoded as an amplitude $A_{{\overset{ˇ}{F}}_{u,v}},\left(u\neq 0,v\neq 0\right)$ in ones complement form. The head parameter of each coefficient is entropy encoded, as discussed below.

Step 5. Entropy Encoding

In the previous step, the quantized DCT coefficients are represented in such a way that they can be efficiently compressed with an entropy encoder such as the Huffman encoder. The Huffman encoding scheme assigns a variable length code (VLC) to each symbol based on its probability. The main idea of VLC is to assign shorter codes to the most probable symbols and longer codes to the less probable symbols. During decompression, a Huffman decoder along with the coding tables are used to recover the symbols from the compressed bitstream.

## 4. Block-Based Compressible Perceptual Encryption Methods

The main idea of the CPE methods is to divide an image into blocks, as discussed in Section 3.2, and perform some geometric and color transformations on them in order to protect the image global contents. Such block-level processing preserves the image local contents such as the spatial correlation of the neighboring pixels within a block. This correlation can be exploited by an image compression algorithm to compress the cipher images. A careful consideration of the block size is required to achieve the best tradeoff between the compression and encryption efficiencies. For example, in the JPEG standard, the smallest allowable block sizes are 16 × 16 and 8 × 8 for color and grayscale image compression, respectively. In general, CPE methods consist of the following three steps:Step 1. Input image representation

An input color image I, whose dimensions are specified by H rows, W columns, and C components, can either be represented as a true color image $I_{H,W,C}$ or a pseudo-grayscale image by concatenating the color components in either the vertical direction as $I_{\left(H\times C\right),W}$ or the horizontal direction as $I_{H,\left(W\times C\right)}$. On the other hand, when the input is a grayscale image $I_{W,H}$, this step is omitted.

Step 2. Block-based encryption

CPE methods perform geometric transformations to change block positions (block permutation) and block orientations (block rotations and inversions), and color transformations (color channel shuffles and negative–positive transformations) to alter pixel values in the blocks. Each of the transformation functions is controlled by a randomly generated key. The set of all these keys serves as the secret key of the CPE scheme. The encryption algorithm of the CPE schemes is a symmetric-key algorithm, where the same set of keys is used for both the encryption of plain images and the decryption of cipher images. The encryption and decryption processes are shown in Figure 3, where $K_{i}$ is the secret symmetric key used in the *i*th step.

Step 3. Compression

The final step is to compress the cipher image using the JPEG image standard. The JPEG color or grayscale image compression mode is chosen based on the input image representation in Step 1.

The PE methods can be classified into two categories based on their preprocessing step: methods that represent the input as a color image and methods that represent the input as a pseudo-grayscale image. The basic form of the first category is to process each color component with the same key; we named these Color CPE methods. These methods can be extended to process each color component independently (Extended CPE) and to introduce sub-block-level processing (IIB–CPE). The second category, where the input is represented in grayscale, is named PGS–CPE methods.

### 4.1. Color CPE Methods

A Color CPE algorithm was proposed in [30,44] for SNS and CPSS applications. In the algorithm, an image $I_{H,W,C}$ with H×W pixels in C=3 color channels is divided into L×M blocks, where L=H/N and M=W/N. A cipher image can be generated as shown in Figure 4, and the procedure described is below:

Step 1. Input image representation

An input color image I, whose dimensions are specified by H rows, W columns, and C components, is represented as a true color image IH,W,C in the RGB colorspace.

Step 2. Block-based encryption

Divide the image $I_{H,W,C}$ into L×M blocks where L=H/N and M=W/N, and each block has C color channels with $N^{2}$ pixels.Shuffle the block positions in the image using a secret key $K_{1}$ generated randomly. The key size is equal to the number of blocks, where each of its entries represent a block’s new position in the scrambled image.Change the block orientations in the shuffled image by a composite function of rotation and inversion transformations. This transformation is controlled by a randomly generated key $K_{2}$ where its entries represent rotation and inversion axis.Change the pixel values by applying a negative–positive transformation function to each pixel in a block randomly chosen by a key $K_{3}$. The $K_{3}$ is a binary key where the elements are uniformly distributed. The negative–positive transformation function for a block B is defined as
(7)$${\overset{´}{p}}_{s,t}=\{\begin{array}{cc} p_{s,t}, & K_{3}{}_{i}=0\\ 255-p_{s,t}, & K_{3}{}_{i}=1 \end{array}$$
where $p_{s,t}$ (s,t=1,⋯,N) is a pixel value in the block and ${\overset{´}{p}}_{s,t}$ is its modified value, and $K_{3}{}_{i}$ is the ith element of the key $K_{3}$.Shuffle the color components of each block using key $K_{4}$. Each element of the $K_{4}$ represents a unique permutation of the color channels.

Step 3. Compression

The final step is to JPEG compress the cipher image obtained in the previous step. Because the input was represented as a color image in the RGB colorspace (Step 1), the JPEG compression can be carried out in the color mode either using RGB or YCbCr colorspace. When a suitable block size is used during encryption, such as N=16, then a user can benefit from the JPEG chroma subsampling for additional compression savings.

### 4.2. Extended CPE Methods

An extension of Color CPE method is proposed in [31,47] to better alter the color distribution. The principal idea is to process each color component independently. The Extended CPE methods can be implemented using the same steps as described in Section 4.1. The main difference between the Color CPE and Extended CPE methods lies in the encryption keys. In Color CPE methods, the same keys are used to encrypt the color components of the image, such as $K_{i}=\left\{K_{i}^{R},K_{i}^{G},K_{i}^{B}\right\}$ where $K_{i}^{R}=K_{i}^{G}=K_{i}^{B}$ and i={1,2,3}. However, in the Extended CPE methods, the encryption keys used in each color component are different, such as $K_{i}=\left\{K_{i}^{R},K_{i}^{G},K_{i}^{B}\right\}$ where $K_{i}^{R}\neq K_{i}^{G}\neq K_{i}^{B}$. Because of this independent processing, the spatial information in each color channel is modified differently, as shown in Figure 5.

In addition, the JPEG compression can be carried out in the color mode as the input was represented as a color image. However, because of the independent color component, the process of the compression of the cipher image should be carried out in a lossless mode, such as in RGB colorspace and without chroma subsampling.

### 4.3. IIB–CPE Methods

An IIB–CPE scheme is proposed in [34,49,50] to expand the keyspace of Color CPE methods. The core idea is to perform sub-block processing. A cipher image can be generated as illustrated in Figure 6, and the procedure is described below:

Step 1. Input image representation

An input color image I, whose dimensions are specified by H rows, W columns, and C components, is represented as a true color image IH,W,C in the RGB colorspace.

Step 2. Block-based encryption

Divide the image $I_{H,W,C}$ into L×M blocks.Perform inside-out transformation on each block. It is carried out in two steps: First, each block is divided into sub-blocks, and then, each sub-block orientation is changed. For example, a block $B_{N,N}$ can be divided into SL×SL sub-blocks, where SL=N/SN, and each sub-block has $SN^{2}$ pixels. Change the sub-block orientations in a given block by a composite function of rotation and inversion transformations by using a random key $K_{1}$.Shuffle the whole block position in the image using a randomly generated secret key $K_{2}$.Change the pixel values by applying a negative–positive transformation function to each pixel in a block randomly chosen using a random key $K_{3},$ as in Equation (7).Shuffle the color components of each block using key $K_{4}$. Each element of the $K_{4}$ represents a unique permutation of the color channels.

Step 3. Compression

The final step is to JPEG compress the cipher image obtained in the previous step. Because the input was represented as a color image in the RGB colorspace (Step 1), the JPEG compression can be carried out in the color mode.

### 4.4. PGS–CPE Methods

A PGS–CPE scheme is proposed in [32,33,48] to deal with format compatibility and chroma-subsampling issues in color-based CPE methods. The principal idea is to represent the input color image in a pseudo-grayscale form in order to benefit from the allowable smallest block size in the JPEG standard for better encryption efficiency. A cipher image can be generated as illustrated in Figure 7, and the procedure is described below:

Step 1. Input image representation

An input color image I in the RGB colorspace, whose dimensions are specified by H rows, W columns, and C components $I_{H,W,C}$, is converted into YCbCr colorspace. The three components $Y_{H,W},$ $Cb_{H,W},$ and $Cr_{H,W}$ are concatenated either in a horizontal direction to form an image $I_{H,\left(C\times W\right)}$ or a vertical direction to form an image $I_{\left(C\times H\right),W}$, as shown in Figure 8. However, for the color-subsampling function (for example, a ratio of 4:2:0), the chroma components are downsampled as $\overset{´}{Cb}=Cb_{H/2,W/2}$ and $\overset{´}{Cr}=Cr_{H/2,W/2}$. The three components $Y_{H,W},$ ${\overset{´}{Cb}}_{H/2,W/2},$ and ${\overset{´}{Cr}}_{H/2,W/2}$ are concatenated either in a horizontal direction to form an image $I_{H,\left(C\times \left(W/2\right)\right)}$ or a vertical direction to form an image $I_{\left(C\times \left(H/2\right)\right),W}$. Here, we assumed that the input image $I_{H,W,C}$ is represented in pseudo-grayscale form without the chroma subsampling as $I_{H,\left(C\times W\right)}$.

Step 2. Block-based encryption

Divide the image $I_{H,\left(C\times W\right)}$ into L×M blocks where L=H/N and M=(C×W)/N, and each block has $N^{2}$ pixels.Shuffle the block positions in the image using a secret key $K_{1}$ generated randomly.Change the block orientations in the shuffled image by a composite function of rotation and inversion transformations. This transformation is controlled by a randomly generated key $K_{2}$.Change the pixel values by applying a negative–positive transformation function to each pixel in a block chosen using a random key $K_{3},$ as in Equation (7).

Step 3. Compression

The final step is to JPEG compress the cipher image obtained in the previous step. Because the input was represented as a grayscale image, the JPEG compression can be carried out in the grayscale mode by using either the luminance or chrominance standard table in the quantization step.

### 4.5. Extension to Grayscale Image Processing

Besides color image encryption and compression, the CPE methods presented above can also be used with grayscale images. A grayscale image consists of only one component as opposed to a color image which has three components. The CPE methods consist of the following two steps for grayscale image encryption and compression:Step 1. Block-based encryption

The CPE methods perform geometric transformations to change block positions (block permutations) and orientations (block rotations and inversions), and intensity transformation (negative–positive transformation) to alter pixel values.

Step 2: Compression

The final step is to compress the cipher image using the JPEG image standard in the grayscale mode either using the standard luminance or chrominance quantization tables.

For the grayscale input, the image representation step is omitted (Step 1 in Section 4) and the PE methods can be classified as methods that transform an entire block (GS–CPE) and methods that incorporate sub-block processing (GS–IIB–CPE). The methods Color CPE, Extended CPE, and PGS–CPE are of class GS–CPE and IIB–CPE is of class GS–IIB–CPE. The following subsections provide an overview of these methods.

#### 4.5.1. GS–CPE

A cipher image can be generated by following the procedure described below:Step 1. Block-based encryption

Divide the grayscale image $I_{H,W}$ into L×M blocks where L=H/N and M=W/N, and each block has $N^{2}$ pixels.Shuffle the block positions in the image using a secret key $K_{1}$ generated randomly.Change the block orientations in the shuffled image by a composite function of rotation and inversion transformations. This transformation is controlled by a randomly generated key $K_{2}$.Change the pixel values by applying a negative–positive transformation function to each pixel in a block randomly chosen using a random key $K_{3},$ as in Equation (7).

Step 2. Compression

The final step is to JPEG compress the cipher image obtained in the previous step. Because the input image is a grayscale image, the JPEG compression is carried out in the grayscale mode with either of the standard quantization tables.

#### 4.5.2. GS–IIB–CPE

A cipher image can be generated by following the procedure described below:Step 1. Block-based encryption

Divide the grayscale image $I_{H,W}$ into L×M blocks where L=H/N and M=W/N, and each block has $N^{2}$ pixels.Perform inside-out transformation on each block. Divide each block into sub-blocks and then change the orientation of each sub-block. For example, a block $B_{N,N}$ can be divided into SL×SL sub-blocks where SL=N/SN and each sub-block has $SN^{2}$ pixels. Change the sub-block orientations in a given block by a composite function of rotation and inversion transformations with a random key $K_{1}$.Shuffle the whole block position using a secret key $K_{2}$ generated randomly.Change the pixel values by applying a negative–positive transformation function to each pixel in a block randomly chosen by using a random key $K_{3,}$ as in Equation (7).

Step 2. Compression

The final step is to JPEG compress the cipher image obtained in the previous step. Because the input image is a grayscale image, the JPEG compression is carried out in the grayscale mode with either of the standard quantization tables.

### 4.6. CPE Encryption Level

For multimedia applications where the security requirement is flexible, the encryption level of the CPE schemes described in Section 4.1, Section 4.2, Section 4.3, Section 4.4 and Section 4.5 can be adjusted accordingly. This can be achieved by performing the CPE steps on selected blocks. For example, to preserve the global contents of the plain image during encryption, the block permutations can be applied selectively to certain blocks of the image. Similarly, the composite function of rotation and inversion, negative–positive transformation function, and color-channel shuffling function can be set as identity functions for the selected blocks to preserve the local contents of the image on a block level.

## 5. Performance Analysis of CPE Schemes

This section presents a comparison between different CPE methods in terms of compression savings and encryption efficiency. In the simulations, compression analyses were carried out on two datasets: the Tecnick sampling dataset [74], which consists of 120 true color images of 1200 × 1200 resolution, and the Shenzhen chest X–ray images dataset [75], which consists of 400 grayscale images of 2048 × 2048 resolution. The CPE methods described in Section 4 were custom implemented due to the unavailability of standard source code, and the JPEG implementation available in [76] was used. Throughout the experiments, the JPEG quality factor qf∈{71,72,⋯,100} was used. In addition, to analyze the CPE methods under various conditions, Table 5 and Table 6 summarize the setup of each method for color and grayscale image compression, respectively.

For the encryption efficiency analysis, the experiments were conducted on the USC–SIPI Miscellaneous dataset [77]. In total, 24 color images were selected from the dataset, uniformly distributed between 256 × 256, 512 × 512, and 1024 × 1024 resolutions.

### 5.1. Visual Analysis

Figure 9a shows an example image from the Tecnick dataset and its cipher images (b–g) obtained from the Color CPE, PGS–CPE, Extended CPE, and IIB–CPE schemes. For visual analysis, the square bounded area in each image is zoomed in and shown below its corresponding image. It can be seen that the global contents of the image are scrambled. Owing to the smaller block sizes, the PGS–CPE achieved better visual encryption of the local details. The cipher images were compressed using the JPEG algorithm without chroma subsampling under different quality factors, and their corresponding recovered images are shown in Figure 9h–ab. During compression, the quality factor was set to qf=71 in Figure 9h–n, qf=85 in Figure 9o–u, and qf=100 in Figure 9v–ab. The images recovered from the cipher images have the same visual appearance as the recovered plain images.

To analyze the chroma subsampling effect, the plain and cipher images given in Figure 10a,b,d–g were compressed with the JPEG algorithm using the chroma subsampling function, as shown in Figure 10. The JPEG algorithm performs chroma subsampling on a block size of 16 × 16. Therefore, when a smaller block size is used in the CPE methods, the downsampled color blocks have pixels from different blocks, wherein the correlation value is low. Interpolating these pixels to recover the original image resolutions results in block artifacts. This effect can be seen in the case of Color CPE (8 × 8) and the IIB–CPE methods shown in Figure 10. In the PGS–CPE, these block artifacts are avoided as the chroma subsampling is completed before the encryption.

For the grayscale image visual analysis, Figure 11 shows an example image from the USC–SIPI dataset and its cipher images (b–d) obtained from GS–CPE and GS–IIB–CPE methods. For visual analysis, the square bounded area in each image is zoomed in and shown below its corresponding image. It can be seen that the global contents of the image are scrambled. Owing to the sub-block processing, the GS–IIB–CPE method achieved better visual encryption of the local details. The cipher images were compressed using the JPEG algorithm under different quality factors, and their corresponding recovered images are shown in Figure 11e–p. During compression, the quality factor was set to qf=71 in Figure 11e–h, qf=85 in Figure 11i–l, and qf=100 in Figure 11m–p. The images recovered from the cipher images have the same visual appearance as the recovered plain images.

### 5.2. Compression Analysis

#### 5.2.1. CPE Compressibility—Energy Compaction Analysis

One of the main steps in the JPEG compression standard is the DCT function, which represents the image in such a way that more compression savings can be achieved in the later steps. For the DC coefficient (u,v=0), Equation (3) can be simplified as
(8)$$F_{\left(0,0\right)}=\frac{1}{N}\overset{7}{\underset{i=0}{\sum }}\overset{7}{\underset{j=0}{\sum }}B_{i,j}.$$
The $F_{\left(0,0\right)}$ is the average value of pixels in a given block, which makes the DC coefficient value independent of the pixel positions. Therefore, the CPE processing steps, such as rotation and inversion, and color-channel shuffle steps have no effect on the DC value. The permutation and negative–positive inversion steps have a smaller effect on the DPCM efficiency. An alternative method to compute the DCT function over each image block is to precompute the basis function points and multiply them with each block as
(9)$$D={TBT}^{\prime },$$
where B represents the image block and T represents the DCT matrix calculated as
$$T_{\left(i,j\right)}=\{\begin{array}{cc} \frac{1}{\sqrt{8}} & if\;i=0\\ \sqrt{\frac{1}{4}}\cos\left[\frac{\left(2j+1\right)i\pi }{16}\right] & if\;i>0 \end{array},$$
where the T multiplication on the left transforms the rows of B, and $T^{\prime }$ multiplication on the right transforms the columns of B. Following the matrix multiplication convention presented in [49], the first product P=TB is a linear combination of the columns of matrix T with weights given by the columns of matrix B. The matrix B with 8 columns and 8 rows can be represented in a compact form as $B=\left[b_{0},\cdots ,b_{7}\right]$, where $b_{i}=\left[b_{0,i},\cdots ,b_{7,i}\right]$ is the *i*th column. The product P is calculated as $P=\left[Tb_{0},\cdots ,Tb_{7}\right]$ where its *i*th column is $P_{i}=Tb_{i}$ which is calculated as
(10)$$P_{i}=\overset{7}{\underset{k=0}{\sum }}T_{*k}b_{ki},$$
which defines a relation between the product matrix elements with respect to the weight matrix. One relation is that changing the entire block orientation (as in Color CPE method) changes only the correlation direction; therefore, the resulting DCT coefficient matrix has the same values but in different positions. On the other hand, when a block symmetry is altered because of the sub-block processing (as in IIB–CPE method), then the coefficient values change as well. For a better understanding of the energy compaction analysis, we extracted two 8 × 8 blocks from the standard Lena image, and both blocks have different correlation coefficients. In the first image block, the horizontal correlation factor is $\sigma_{h}=0.95$ and the vertical correlation factor is $\sigma_{v}=0.96$, whereas in the second image block, the horizontal correlation factor is $\sigma_{h}=0.49$ and the vertical correlation factor is $\sigma_{v}=0.52$. The DCT transformation of the original and scrambled image blocks are shown in Figure 12a–d and Figure 12e–h, respectively. The scrambled images in Figure 12b,f were obtained by changing the entire block orientation (that is rotation by 90°). The scrambled images in Figure 12c,d,g,h were obtained by dividing the blocks into sub-blocks and then changing the orientations of the sub-blocks randomly. In this example, one sub-block was rotated by 90° and one sub-block was flipped over the vertical axis. It can be seen in Figure 12b,f that because of the entire block transformation, the DCT coefficient values remain the same, and only their positions change. The DCT matrix obtained is equivalent to the diagonal flip of the original matrix. On the other hand, the sub-block processing changed the DCT coefficient values, as shown in Figure 12c,d,g,h. Nonetheless, the JPEG quantization step significantly reduced the difference in the DCT coefficients of the original and transformed image blocks, as shown in Figure 12. In the quantization step, the standard luminance quantization table with qf=80 was used. In fact, during intermediate encoding, the zigzag scan of the DCT matrix resulted in almost the same number of zero AC coefficients which can be encoded as the JPEG EOB identifier in the same manner in all of the cases, as described in Section 3.3.

#### 5.2.2. CPE Compression—Efficiency Analysis

For compression analysis, Figure 13, Figure 14, Figure 15, Figure 16, Figure 17, Figure 18 and Figure 19 show the RD curves according to the setups described in Table 5 and Table 6. In each plot, the x-axis is the compression savings in terms of bitrate and the y-axis is the recovered image quality represented as an MS–SSIM measure value in dB. The RD curves were quantitatively compared by using the BD difference measures proposed in [78]. For an equivalent quality, the BD rate gives the difference between two bitrates in percentage, and for the equivalent bandwidth, the BD quality gives the average dB difference between RD curves. Following [49], the BD rate difference is calculated for the MS–SSIM measure instead of the PSNR, and the value of MS–SSIM (M) is $-10\log_{10}\left(1-M\right)$.

##### JPEG Plain Image Compression

The JPEG algorithm can be implemented for the compression of color and grayscale images, as described in Section 3.3. For color image compression (without chroma subsampling), an input can be represented either in the RGB or YCbCr colorspace. However, when subsampling is to be utilized, then it is necessary to represent the image in the YCbCr colorspace. Unlike color images, a grayscale image consists of only one component; therefore, the colorspace conversion step is omitted, and in the quantization step, either of the standard luminance (Table 1) or chrominance (Table 2) tables can be used.

In the JPEG standard, an input color image is represented in the YCbCr colorspace for better compression savings. Though this colorspace conversion is a lossless function, rounding off its output values to the nearest integers introduces some information loss. Therefore, the YCbCr input representation (M11) traded the image quality for better savings compared to the RGB colorspace (M10), as shown in Figure 13. According to the BD-rate measure shown in Figure 13 (M10 vs. M11), M11 required 8% more bitrate for the equivalent quality images of M10.

A color image can be represented as a pseudo-grayscale image by concatenating its three components in either of the horizontal or vertical direction, as discussed in Section 4.4. This pseudo-grayscale representation is the basic principle for PGS–PE methods to achieve encryption efficiency. Therefore, in our analysis, we have also considered the comparison of the JPEG compression efficiency on color and pseudo-grayscale representation of the input images. For this purpose, the input was first converted to pseudo-grayscale representation, and then, the resulting image was compressed with the JPEG algorithm in the grayscale mode. Because either of the luminance or chrominance quantization tables can be used, the JPEG performance was compared on both tables. The images compressed in grayscale mode (M13 and M14) followed the same trend as color image compression in the YCbCr colorspace (M11), as shown in Figure 13. The image quality was being traded for better bitrate. According to the BD-rate measure shown in Figure 13, when the images were compressed in grayscale with the luminance quantization table (M13), it required 8% more bitrate for the equivalent quality of M10, whereas there was a negligible bitrate difference compared to M11. Similarly, when the chrominance table is used for quantization during compression (M14), then the bitrate difference increased to 13% and 2% compared to color mode compression carried out by M10 and M11, respectively. For the choice of quantization table analysis, the luminance table (M13) provided 2% better bitrate savings compared to the chrominance table (M14).

The analyses discussed so far are for the JPEG compression without chroma subsampling function. When the JPEG algorithm is implemented with chroma subsampling, then it is necessary to represent the input image in the YCbCr colorspace. Therefore, the only analysis was to compare the compression in color and grayscale mode. The pseudo-grayscale representations of the input images were obtained as discussed in Section 4.4. Because the images are in YCbCr colorspace in both the color and pseudo-grayscale representations, they followed the same trend as in Figure 13. In the lower bitrate region, the grayscale mode (M15 and M16) had better quality than the color mode, whereas in the higher bitrate region, the trend was reversed, as shown in Figure 13. In contrast to the JPEG compression without chroma subsampling, where the color mode delivered better bitrate savings than the grayscale mode, here, the grayscale representation achieved 8% and 6% bitrate savings compared to the color mode (M12) with luminance (M15) and chrominance (M16) quantization tables, respectively. For the choice of quantization table analysis, the luminance table (M15) provided 6% bitrate savings compared to the chrominance table (M16).

##### JPEG Plain versus Cipher Image Compression

We compared the JPEG compression performance on the plain and cipher images. The color perceptual encryption methods (Color CPE, Extended CPE, and IIB–CPE methods) encrypt the images in the RGB colorspace, and their compression can be carried out in either the RGB or YCbCr colorspace. Therefore, we compared the JPEG compression without chroma subsampling of the plain and PE cipher images in both colorspaces, as shown in Figure 14. It is important to note that the Extended CPE disrupt the spatial information in each color channel, which makes them unsuitable for compression in the YCbCr colorspace and with chroma subsampling; therefore, we have omitted them from this analysis. In both colorspaces, the compression of the cipher images without chroma subsampling followed almost the same trend as that of the plain image compression, as shown in Figure 14. Specifically, according to the BD-measures in Figure 14b,d, the bitrate difference was 3% and 5% for the RGB (M10 vs. {M1, M17, M19, M21}) and YCbCr (M11 vs. {M2, M22}) colorspaces across all encryption methods, respectively. On the other hand, when the compression was carried out with chroma subsampling, as shown in Figure 15, the bitrate difference is 6% for the Color CPE method (M3), and for the methods that incorporate sub-block processing (M27), the compression savings drastically decreased, i.e., a 112% bitrate difference.

The grayscale PE method (PGS–CPE method) has a preprocessing step of representing the input as a pseudo-grayscale image by concatenating its three components along the horizontal or vertical direction, as discussed in Section 4.4. As suggested in the literature, the input is first converted into the YCbCr colorspace before any preprocessing. For a fair comparison, the analysis is presented for both the color (YCbCr colorspace) and grayscale compression modes of the plain images. Throughout the experiments, the two IJG standard tables were used during the quantization step. In addition, custom quantization tables provided in [48] were used only for the compression of PGS–CPE cipher images.

When the JPEG algorithm is implemented without the chroma subsampling function, the plain image compression (M11, M13, and M14) had a better MS–SSIM RD curve compared to the compression of PGS–CPE images (M4, M5, and M6), as shown in Figure 16. The minimum bitrate difference of 10% was achieved with the luminance quantization table (M13 vs. M4 and M11 vs. M14), as shown in Figure 16. In addition, no performance efficiency was gained when using the custom quantization table compared to the standard luminance table. However, compared to the chrominance table, a 3% better bitrate was achieved.

On the other hand, when the compression is carried out with chroma subsampling, the PGS–CPE methods (M7 and M9) have a better RD curve than the color plain image compression (M12), as shown in Figure 16, i.e., the methods M7 and M9 require a lesser bitrate than the color plain images. The reason is that for color image compression, the JPEG standard uses two quantization tables, such as luminance and chrominance tables. Quantization with the chrominance table results in more information loss than with the luminance or the custom table proposed in [48]. This observation can be supported by M12 vs. M8 in Figure 16, where both the luminance and color components were quantized by the chrominance table, and 2% more bitrate was required to achieve equivalent image quality. It was observed in the earlier analysis that the JPEG efficiency improved with the pseudo-grayscale representation; therefore, we compared the JEPG performance on the grayscale representation of plain images (M15 and M16) and the cipher images (M7, M8 and M9). Figure 16 shows that M15 and M16 have a better RD curve than the PGS–CPE methods. Specifically, when the compression was performed on the grayscale representation of both plain and cipher images, there was 8% minimum and 10% maximum datarate difference in the case of luminance and chrominance tables, respectively. The efficiency gain when using the custom quantization table remains the same as in the case of compression without chroma subsampling.

In our simulations, the final analysis for color image compression compared the sub-block size effect on the JPEG compression efficiency. When the sub-block size is chosen to be smaller than the one allowed in the JPEG standard, there is a significant difference in the RD curves as shown in Figure 17a–c for the JPEG compression without and with chroma subsampling, respectively. For the JPEG compression without chroma subsampling, the datarate difference increased as the sub-block size decreased in both colorspaces, as shown in Figure 17. Overall, the maximum datarate difference is 78% and 82% for the smallest sub-block size in the RGB (M25) and YCbCr (M26) colorspaces, respectively. On the other hand, when chroma subsampling is implemented, the datarate difference has an inverse relation with the sub-block size because the use of smaller sub-block sizes better preserves the correlation within a block [49]. The maximum bitrate difference was 105% and the minimum bitrate difference was 61% for the M27 and M29 methods, respectively.

##### Grayscale Image Compression Analysis

Quantization tables analysis: For grayscale image compression, the JPEG standard provides two standard quantization tables: the luminance and chrominance quantization tables, as given in Table 1 and Table 2, respectively. This subsection compares the JPEG performance with the choice of quantization tables, as shown in Figure 18. In both cases, for plain and encrypted image compression, the choice of the quantization table has a negligible effect on the performance of the JPEG algorithm. Overall, the maximum datarate difference is below 1.5%, whereas quality difference is below 0.2 dB.

Compression of plain images versus encrypted images: This subsection presents the JPEG compression performance on the plain and PE cipher images, as shown in Figure 19. The cipher images were obtained from the encryption methods that transform an entire block (G1 and G2) and methods that incorporate sub-block processing (G3–G6). The analyses were carried out for the two standard quantization tables. Specifically, the JPEG algorithm was implemented with the luminance table in methods G1, G3, G5, and G7 and thr chrominance table in methods G2, G4, G6, and G8, as given in Table 6. Compared to the compression of the plain images (G8), the cipher image compression requires 5% (G2) and 12% (G4 and G6) more bitrate, whereas the quality degradation is negligible. On the other hand, when using the chrominance table in the quantization step, the datarate difference increased by 3% at maximum for the compression of the PE images compared to the plain image compression (G7). Overall, the methods that incorporate sub-block processing penalized the JPEG algorithm more than the methods that process an entire block.

### 5.3. Encryption Analysis

#### 5.3.1. Correlation Analysis

An encryption algorithm should eliminate correlation among adjacent pixels in an image for better security. In general, the correlation coefficient ρ(x,y) between two distributions x and y each with N elements is given by
(11)$$\{\begin{array}{c} \rho \left(x,y\right)=\frac{1}{N}\overset{N}{\underset{i=1}{\sum }}\left(\frac{x_{i}-\mu_{x}}{\sigma_{x}}\right)\left(\frac{y_{i}-\mu_{y}}{\sigma_{y}}\right)\\ \mu_{a}=\frac{1}{N}\overset{N}{\underset{i=1}{\sum }}a_{i}\\ \sigma_{a}=\sqrt{\frac{1}{N}\overset{N}{\underset{i=1}{\sum }}{\left|a_{i}-\mu \right|}^{2}} \end{array}$$
For the coefficient ρ∈{−1.0, 1.0}, ρ=0 shows that there is no correlation, ρ<0 shows negative correlation, and ρ>0 shows positive correlation. The negative correlation means that when one value is increasing, the other is decreasing, and the positive correlation means that both values are either increasing or decreasing. For the correlation analysis, we have performed two experiments. First, we have shown the correlation between adjacent pixels randomly chosen from the whole image. The encryption algorithms are block-based; therefore, the correlation among the neighboring pixels was still high in the cipher images, as shown in Table 7. In order to preserve the JPEG compression performance efficiency on the cipher images, the correlation in the block of at least 8×8 in size should not be altered. At first, it may seem like the CPE algorithms are vulnerable, as also mentioned in [5]; therefore, in the second experiment, we have analyzed the correlation among adjacent blocks by taking the pixels on the borders only. It can be seen that on a block level, the cipher image had low correlation and exhibits favorable encryption properties. Table 7 presents the correlation analysis for the entire dataset in diagonal, horizontal, and vertical directions for plain images and CPE cipher images.

#### 5.3.2. Histogram Analysis

The histogram of an image gives the intensity distribution as the number of pixels at each intensity level. For a plain image, the histogram is a skewed distribution concentrated at one location, and a cipher image has a uniform distribution. To quantify the characteristics of a histogram R, histogram variance V(R) is calculated as
(12)$$\{\begin{array}{c} V\left(R\right)=\frac{\sum_{i=1}^{N}{\left(R_{i}-\mu_{R}\right)}^{2}}{N-1}\\ \mu_{R}=\frac{1}{N}\overset{N}{\underset{i=1}{\sum }}R_{i} \end{array},$$
where N is the level of intensities in the image and μ is the mean of the image histogram. A small value of V(R) means a uniform distribution. Table 7 shows the mean V(R) values across the whole dataset for plain and cipher images. In all cases, the V(R) values of cipher images are smaller than those of the plain images; therefore, this reduces the information characteristics of the image. The PGS–CPE has the greatest V(R) value among the evaluated methods.

#### 5.3.3. Information Entropy Analysis

The information entropy shows the degree of randomness in an image. The entropy of an image H(I) is given by
(13)$$H(I)=-\overset{M}{\underset{i=1}{\sum }}p_{i}\log_{2}(p_{i}),$$
where $p_{i}$ is the probability of a pixel value in the image. For a truly random image with N=256 intensity levels, the ideal value of the entropy should be closer to $H\left(I\right)=log_{2}\left(N\right)=8.$ Table 7 shows the mean of entropy values across the whole dataset for plain and cipher images. The entropy values are smaller than the ideal value of H(I)=8 because the PE methods preserve the image contents on a block level. Nonetheless, H(I) values of cipher images were greater than those of the plain images; therefore, this resulted in better randomness. In addition, PGS–CPE methods have the smallest H(I) value among the evaluated CPE methods.

#### 5.3.4. Differential Attack Analysis

In order to be resistant against differential attack, an encryption algorithm should have the ability to generate two different cipher images for plain images with a minor difference. The degree of change can be quantified by two metrics, namely, the number of pixels change rate (NPCR) and the unified average changing intensity (UACI). The NPCR gives the percentage difference between two cipher images and the UACI gives the average intensity of differences between the two images. For this purpose, a plain image $I_{1}$ of size M is slightly modified by randomly changing one of its pixel values to generate another image, $I_{2}$. The two plain images $I_{1}$ and $I_{2}$ are encrypted using the same encryption key to obtain the cipher images $C_{1}$ and $C_{2}$, respectively. The NPCR and UACI parameters are calculated for the cipher images $C_{1}$ and $C_{2}$ as
(14)$$\begin{array}{c} NPCR=\frac{\sum_{i,j}D_{i,j}}{M}\times 100\%,\\ D_{i,j}=\{\begin{array}{cc} 1, & C_{1}\left(i,j\right)\neq C_{2}\left(i,j\right)\\ 0, & C_{1}\left(i,j\right)=C_{2}\left(i,j\right) \end{array}. \end{array}$$
(15)$$UACI=\frac{1}{M}\left[\sum_{i,j}\frac{\left|C_{1}\left(i,j\right)-C_{2}\left(i,j\right)\right|}{255}\right]\times 100\%.$$
For $C_{1}$ and $C_{2}$ to have the ideal values of NPCR and UACI, the minor change in the plain images should be reflected across the whole cipher images. Usually, the diffusion process, which makes the current ciphertext dependent on the previous ones, achieves this property. However, in the CPE schemes, there is no such operation. In fact, the only step that changes pixel values is the negative–positive transformation function, where 50% of the blocks or pixels are randomly XORed with 255. As a result, the CPE schemes may be vulnerable to differential attacks. Nonetheless, the use of different keys for each image, as suggested in the literature, provides a certain level of resistance against the attack.

#### 5.3.5. Jigsaw Puzzle Solver (JPS) Attack Analysis

The CPE schemes perform block-based encryption processes, and their resulting cipher images preserve the intrinsic properties of the original image; therefore, it is necessary to evaluate their robustness against JPS attack, as proposed in [35] and its extended version to accommodate the sub-block processing proposed in [49]. The JPS is a cipher-text only attack, where each block of the cipher image can be treated as a piece of a jigsaw puzzle. The goal is to reconstruct the plain image fully or partially from the cipher image. Robustness against the attack can be quantified by using the following three measures [79,80]:

Direct comparison ($D_{c}$) estimates the ratio of the blocks that are in correct positions in the recovered image as they would have been in the original image. Let I be the original image, $I_{r}$ the recovered image, $p_{i}$ is the ith piece, and n is the total number of pieces; then, $D_{c}\left(I_{r}\right)$ is given by
(16)$$\{\begin{array}{c} D_{c}\left(I_{r}\right)=\frac{1}{n}\overset{n}{\underset{i=1}{\sum }}d_{c}\left(p_{i}\right),\\ d_{c}\left(p_{i}\right)=\{\begin{array}{cc} 1, & I_{r}\left(p_{i}\right)=I\left(p_{i}\right)\\ 0, & I_{r}\left(p_{i}\right)\neq I\left(p_{i}\right) \end{array} \end{array}$$

Neighbor comparison ($N_{c}$) estimates the ratio of adjacent neighboring blocks that are correctly joined. For the recovered image $I_{r}$ with B boundaries among the pieces, and $b_{i}$ is the ith boundary, then $N_{c}\left(I_{r}\right)$ is given by
(17)$$\{\begin{array}{c} N_{c}\left(I_{r}\right)=\frac{1}{B}\overset{B}{\underset{i=1}{\sum }}n_{c}\left(b_{i}\right),\\ n_{c}\left(b_{i}\right)=\{\begin{array}{cc} 1, & \mathrm{if}\;b_{i}\;\mathrm{is}\;\mathrm{joined}\;\mathrm{correctly}\\ 0, & Otherwise \end{array} \end{array}$$

Largest component comparison ($L_{c}$) estimates the ratio of the largest joined blocks that have correct neighbor adjacencies with other blocks in the component. For the recovered image $I_{r}$ with n partial correctly assembled areas and the number of blocks in the ith assembled area, $L_{c}\left(I_{r}\right)$ is given by
(18)$$L_{c}\left(I_{r}\right)=\frac{1}{n}\underset{i}{\max}\left\{l_{c}\left(I_{r},i\right)\right\}$$

The measures score $D_{c},N_{c},L_{c}\in \left\{0,\;1\right\},$ with 1 being the highest assembled score. Table 8 summarizes the robustness of each CPE method against the jigsaw puzzle attack. It is important to note that the measures scores reported here are from their respective papers. The PGS–CPE methods show a better resistance against the JPS attack among the evaluated CPE methods. The main reason for this is the use of smaller block sizes and better scrambling of the color components. The Extended CPE methods have achieved a comparable performance to the PGS–CPE. On the other hand, the IIB–CPE methods have achieved better resistance against the JPS attack than the Color CPE methods, owing to the sub-block processing.

#### 5.3.6. Robustness Analysis

In this section, we analyze the robustness of CPE schemes against the data loss attack and noise attack. Figure 20 shows the original image, and its cipher images in Figure 20b,h were obtained from the Color CPE, Extended CPE, PGS–CPE, and IIB–CPE schemes. For the data loss attack analysis, we have cropped different regions (i.e., setting the pixel values equal to zero) from the cipher image, as shown in Figure 21a,g for the cipher images in Figure 20b,h. Their corresponding recovered images are shown in Figure 21h,n. It can be seen that the images have recovered successfully without the corrupted blocks. In the case of the Color CPE (Figure 21) and IIB–CPE (Figure 21l–n) images, the lost blocks do not have any color because in each channel, blocks from the same locations have been lost, and the white blocks are the result of the negative–positive transformation step. On the other hand, for the Extended CPE (Figure 21) and PGS–CPE (Figure 21) images, the lost blocks are not from the same locations in the color channels; therefore, the missing blocks have color and certain spatial information appears in them. Similarly, for the noise attack analysis, the cipher images (Figure 20b–h) were added with Gaussian noise (Figure 22a–g) and salt–pepper noise (Figure 22o,u). Their corresponding recovered images are shown in Figure 22h–n and 22v–ab, respectively. In the case of Gaussian noise, the recovered images are blurred in comparison to the original images across all CPE methods. For the salt–pepper noise, the noisy pixels of the cipher images were inherited in the recovered image without affecting the rest of the image. For quantitative analysis, Table 9 summarizes the average MS–SSIM of the recovered images across the whole dataset. Overall, the methods that represent input as a color image have better resilience against data loss and noise. The CPE methods are robust against the noise and data loss attacks owing to the lack of the diffusion process.

#### 5.3.7. Keyspace Analysis

In general, the encryption algorithm of the CPE consisted of four secret symmetric keys: $K_{1}$ permutation key, $K_{2}$ rotation and inversion key, $K_{3}$ negative–positive transformation key, and $K_{4}$ color-channel shuffling key. Each key $K_{i},\;i=\left\{1,2,3\right\}$ is a set of three keys, one for each component of the image, and is denoted as $K_{i}=\left\{K_{i}^{R},K_{i}^{G},K_{i}^{B}\right\}$. The keyspace K of a CPE algorithm is the set of all keys used in the encryption steps as $K=\left\{K_{1},K_{2},K_{3},K_{4}\right\}$ and the key size is given by the set cardinality as |K|.

As discussed in Section 4, in the CPE methods, an input color image $I_{W\times H\times C},$ with W×H pixels in C color channels, is grouped into nonoverlapping square blocks with $N^{2}$ pixels. The number of blocks $B_{c}$ in a color channel c is given by
(19)$$B_{c}=L\times M,$$
and the number of blocks B in the image is given by
(20)$$B=3\times B_{c}.$$
When a block B of size N×N pixels is divided into SL×SL smaller blocks of size $SN^{2}$ for sub-block processing, the number of sub-blocks $SB_{c}$ in a color channel c is
(21)$$SB_{c}=\left(SL\times SL\right)\times B_{c},$$
and the number of sub-blocks SB in the image is given by
(22)$$SB=3\times SB_{c}.$$
The keyspace $K_{CC}$ for the Color CPE scheme based on Equation (19) can be derived as
(23)$$K_{CC}=\left\{K_{1,CC},K_{2,CC},K_{3,CC},K_{4,CC}\right\}\;\left|K_{CC}\right|=3\left(B_{c}!\right)\cdot 3\left(8^{B_{c}}\right)\cdot 3\left(2^{B_{c}}\right)\cdot 6^{B_{c}}.$$
Because the Color CPE scheme used the same key for each color component, its keyspace size becomes
(24)$$\left|K_{CC}\right|=B_{c}!\cdot 8^{B_{c}}\cdot 2^{B_{c}}\cdot 6^{B_{c}}.$$
The keyspace $K_{EC}$ for Extended CPE schemes based on Equation (19) can be derived as
(25)$$K_{EC}=\left\{K_{1,EC},K_{2,EC},K_{3,EC},K_{4,EC}\right\}\;\left|K_{EC}\right|=3\left(B_{c}!\right)\cdot 3\left(8^{B_{c}}\right)\cdot 3\left(2^{B_{c}}\right)\cdot 6^{B_{c}}.$$
Here, the keyspace for the first three steps increased by a factor of three as compared to $\left|K_{CC}\right|$ in Equation (24). The reason for this is that the Extended CPE schemes perform the encryption steps independently in each color component. In addition, the color-channel shuffling step scrambles the blocks in the three color components; therefore, Equation (25) can be simplified as
(26)$$\left|K_{EC}\right|=\left(3B_{c}\right)!\cdot 8^{3B_{c}}\cdot 2^{3B_{c}}\;=B!\cdot 8^{B}\cdot 2^{B}.$$
The keyspace $K_{IC}$ for the IIB–CPE can be derived as
(27)$$K_{IC}=\left\{K_{1,IC},K_{2,IC},K_{3,IC},K_{4,IC}\right\}\;\left|K_{IC}\right|=3\left(B_{c}!\right)\cdot \left(3\left(8^{SB_{c}}\right)\cdot 3\left(8^{B_{c}}\right)\right)\cdot 3\left(2^{B_{c}}\right)\cdot 6^{B_{c}}.$$
Similar to the Color CPE scheme, the IIB–CPE uses the same key for each color component, and its keyspace becomes
(28)$$\left|K_{IC}\right|=B_{c}!\cdot \left(8^{SB_{c}}\cdot 8^{B_{c}}\right)\cdot 2^{B_{c}}\cdot 6^{B_{c}}.$$
Compared to $\left|K_{CC}\right|$ in Equation (24), $\left|K_{IC}\right|$ is increased by a factor of $8^{SB_{c}}$ because of the sub-block processing. This increment depends on the sub-block size; specifically, when the number of pixels in a sub-block is $SN^{2}\in \left\{8^{2},4^{2},2^{2}\right\}$, the keyspace size is increased by a factor of $8^{SB_{c}}\in \left\{8^{4B_{c}},8^{16B_{c}},8^{64B_{c}}\right\}$, respectively.

The keyspace $K_{PC}$ for the PGS–CPE scheme can be derived without the last term $K_{4}$ as the methods lack the color-channel shuffling step:(29)$$K_{PC}=\left\{K_{1,PC},K_{2,PC},K_{3,PC}\right\}\;\left|K_{PC}\right|=B!\cdot 8^{B}\cdot 2^{B}.$$
The number of blocks is increased by a factor of three compared to the Color CPE schemes. Similar to Extended CPE methods, the PGS–CPE schemes process each image block independently, as the color channels are concatenated in a single component. In addition, in contrast to the color-based CPE methods, where the smallest block size used is 16 × 16, the PGS–CPE schemes can benefit from the smallest allowable block size in the JPEG standard (8 × 8); the number of blocks are increased four times, and Equation (29) can be modified as
(30)$$\left|K_{PC}\right|=\left(4B\right)!\cdot 8^{\left(4B\right)}\cdot 2^{\left(4B\right)}.$$
Overall, based on Equations (24), (26), (28), and (30), the relation between the keyspace sizes of the CPE methods for color image encryption can be established as
(31)$$\left|K_{PC}\right|\gg \left|K_{EC}\right|\gg \left|K_{IC}\right|>\left|K_{CC}\right|.$$

For the encryption of grayscale images, the CPE consisted of three secret symmetric keys: $K_{1}$ permutation key, $K_{2}$ rotation and inversion key, and $K_{3}$ negative–positive transformation key. The keyspace K of a CPE algorithm for the grayscale image encryption is the set of all keys used in the encryption steps as $K=\left\{K_{1},K_{2},K_{3}\right\}$.

Similar to the encryption of color images, in the CPE methods, an input grayscale image $I_{W\times H},$ with W×H pixels, is divided into nonoverlapping square blocks with $N^{2}$ pixels. The number of blocks B in the image is given by
(32)B=L×M,
and when a block B of size N×N pixels is divided into SL×SL smaller blocks of size $SN^{2}$ for the sub-block processing, the number of sub-blocks SB in the image is given by
(33)SB=(SL×SL)×B.
The keyspace $K_{GC}$ for the GS–CPE schemes based on Equation (31) can be derived as
(34)$$K_{GC}=\left\{K_{1,GC},K_{2,GC},K_{3,GC}\right\}\;\left|K_{GC}\right|=B!\cdot 8^{B}\cdot 2^{B}.$$
The keyspace $K_{GIC}$ for the GS–IIB–CPE can be derived as
(35)$$K_{GIC}=\left\{K_{1,GIC},K_{2,GIC},K_{3,GIC}\right\}\;\left|K_{GIC}\right|=B!\cdot \left(8^{SB}\cdot 8^{B}\right)\cdot 2^{B}.$$
Compared to GS–CPE, where an entire block is transformed, GS–IIB–CPE has a larger keyspace because of the sub-block processing in the rotation and inversion step.

## 6. Perspectives and Future Research Direction

### 6.1. Compression Perspective

The colorspace conversion is lossless in nature; however, the original values cannot be recovered because of its rounding function. Therefore, to achieve the equivalent quality of the images compressed in the RGB colorspace, the JPEG compression in the YCbCr colorspace requires more bitrate. When considering applications such as data-hiding schemes, which have reversibility as the main condition, the JPEG compression should be carried out in the RGB colorspace. However, this does not obsolete the use of the YCbCr colorspace as it is vital to the JPEG chroma subsampling step. In the analysis, it was shown that when using chroma subsampling, the grayscale compression with the luminance quantization table (here, the input color image is represented as a pseudo-grayscale image [48]) is better than the JPEG color mode of compression. This is because, usually, in the color mode, two separate tables are used for the quantization of the luminance and color components, where the chrominance table heavily quantized its corresponding DCT matrices.

When comparing the JPEG compression performance of the CPE methods, the color methods (such as Color CPE, Extended CPE, and IIB–CPE (8 × 8)) have a smaller effect on the JPEG efficiency than that of the grayscale methods (PGS–CPE). The reason for this is that the color methods used a block size of 16 × 16, and during compression, when the image is divided into 8 × 8 blocks, each DC coefficient has one correlated DC coefficient, which results in DPCM encoding efficiency. When using chroma subsampling, the same explanation is valid only for the luminance component.

For the plain grayscale image compression, the choice of quantization table had a negligible effect on the JPEG performance. The reason is that a grayscale image (for example, X-ray images in our analysis) does not correspond to the luminance or chrominance component of the YCbCr colorspace. However, the JPEG algorithm benefited from the luminance quantization table for the compression of the CPE-generated cipher images.

### 6.2. Encryption Perspective

The CPE schemes exhibit properties that are favorable for image encryption, such as randomness, decorrelation, and larger keyspace size. However, one of the main issues with the CPE algorithms is that they are not robust against differential attacks, as discussed in Section 5.3.4. Because the encryption is realized on a block level individually, they have a low diffusion property. In the related literature of CPE methods, a solution to this problem is to use different keys for the encryption of each image. Therefore, if certain secret information is discovered about one image, it will not be useful for another image. This solution is adequate in a scenario where the photo creator and consumer are the same person, such as photo storage applications. However, in photo sharing applications, the key establishment for every photo will result in a communication overhead and waste of computational resources. Similarly, in privacy-preserving applications, the use of different keys may not achieve the desired output. For example, with the recent popularity of CPE-based PPML applications (as in [49,64,65,66]), careful consideration should be given to how the cipher images are generated and whether the use of different keys will affect the model performance.

### 6.3. Security and Usability Perspective

The main reason for adopting a perceptual encryption algorithm instead of another image encryption algorithm is to trade security for usability, as shown in Figure 1. Therefore, the reviewed encryption schemes can be chosen according to a given applications requirements. For example, the PGS–CPE scheme is the most secure one, as given in Equation (31), which makes it the most suitable option for applications such as photo sharing and archiving. For such applications, PGS–CPE-generated cipher images can be efficiently compressed by the JPEG standard with and without the chroma subsampling function. However, when it comes to applications such as reversible data-hiding systems, where there is a strict lossless requirement, or privacy-preserving applications, PGS–CPE schemes are not sufficient. As pointed out in [69], the lossless compression algorithm should be used for reversibility, which makes the YCbCr conversion function and pseudo-grayscale representation of the PGS–CPE unnecessary. One simple solution is to omit these steps, which makes the PGS–CPE methods similar to the Extended CPE methods. Compared to the Color CPE and IIB–CPE methods, the Extended CPE schemes are more suitable for reverse data-hiding applications, owing to their larger keyspace size. However, in the second case of privacy-preserving computation applications, both PGS–CPE and Extended CPE methods are not adequate, as they disrupt the spatial information of the image mainly because they independently perform the blocks permutation step in each color channel. Therefore, the Color CPE and IIB–CPE methods are viable schemes for privacy-preserving applications, as they preserved the image spatial contents. In such applications, preserving the algorithm performance is more important than the compression savings; thus, the JPEG chroma subsampling step is often omitted. Therefore, a smaller block size can be used in the Color CPE schemes, and when more security is desirable, then sub-blocks of smaller sizes can be used in the IIB–CPE schemes.

### 6.4. Future Research Direction

One of the reasons for the JPEG compression efficiency degradation in the CPE generated images is the use of the standard tables in its quantization and entropy encoding steps. These tables were originally designed based on the plain image statistics; therefore, they are not as compatible with the compression of cipher images as they are with the plain images. Nonetheless, the JPEG standard allows the use of user-defined custom tables in these stages. Though the quantization tables proposed in [48] did not achieve the desired efficiency compared to the luminance table, they improved the JPEG performance compared to the chrominance table. This gives an important indication that designing custom tables can reduce the JPEG performance gap. The principle for efficient table design can be defined by the analysis presented in Section 5.2.1. Specifically, it was observed that the encryption algorithm changes the DCT matrix orientation; therefore, the quantization table design should have certain symmetry in order to mitigate this effect, which is missing in the custom tables proposed in [48]. To aid the JPEG algorithm in the compression of CPE images, designing custom tables could be one interesting research direction. In addition, in the reviewed techniques, the PE-based encryption is carried out in such a way that the resulting cipher images are mainly compatible with the JPEG compression algorithm because the JPEG is one of the most widely available image standards on the internet and consumer devices. However, besides the DCT-based compression algorithms, other transformation functions exist, such as the wavelet transform, which are efficient and have better compression performance. Therefore, making the PE algorithms suitable with such compression algorithms could be an interesting approach.

Despite the grayscale representation and sub-block processing, the keyspace size of the CPE algorithms are still constrained by the smallest allowable block size used in the JPEG standard. Therefore, either incorporating the sub-block processing in the PGS–CPE and Extended CPE methods or adopting the pseudo-grayscale representation in the IIB–CPE methods could be an interesting approach for better security. Especially in the latter case, as the chroma subsampling issues will also be resolved.

In recent years, the applications of the CPE methods have been extended to the PPML domain. However, when such applications were considered, the images were lightly compressed, i.e., larger values were used for the JPEG quality factor. The main reason is that, in general, the DL models are not robust against different types of image perturbations, and when they are combined with the encryption, the task of ML algorithms becomes more complex. In this regard, data augmentation techniques that account for the changes in data distribution have been proven efficient. Therefore, developing techniques that can deal with these issues in the encryption domain could be another research direction.

## 7. Conclusions

In this paper, we surveyed the JPEG-compatible block-based perceptual encryption methods. Different CPE schemes were comprehensively analyzed, and their merits were presented in the context of different applications. These schemes were originally designed to meet the dual requirements of image data transmission and storage. Recently, their applications have been extended to computations in the encryption domain, notably, PPML tasks, wherein the requirements differ. Hence, this necessitates careful consideration of the target application demands in the design of CPE schemes. In addition, we identified several potential research directions that can be followed in future studies.

## Figures and Tables

**Figure 1 sensors-23-04057-f001:**
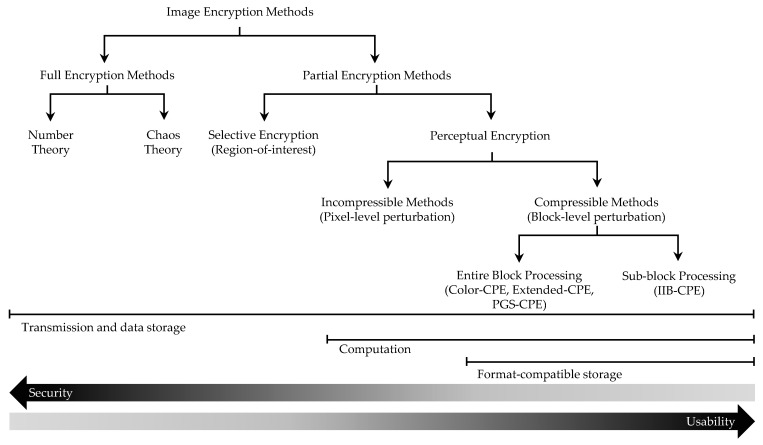
A taxonomy of image encryption methods based on their levels of security. From left to right, the encryption algorithms computational complexity decreases and security is traded for usability, i.e., to enable other multimedia applications such as format compliant storage and even processing the encryption domain.

**Figure 2 sensors-23-04057-f002:**
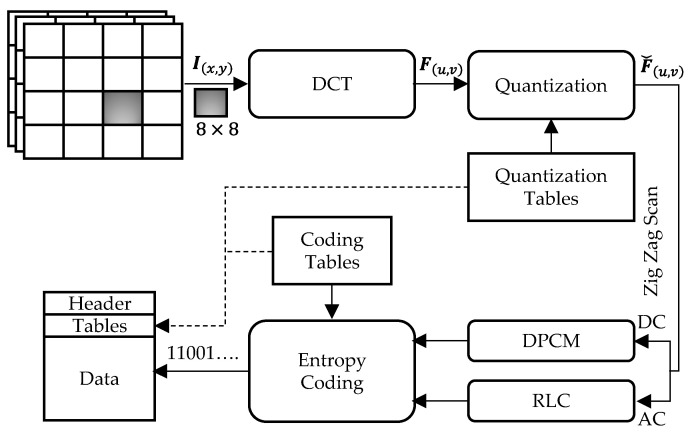
An illustration of the JPEG image compression algorithm.

**Figure 3 sensors-23-04057-f003:**
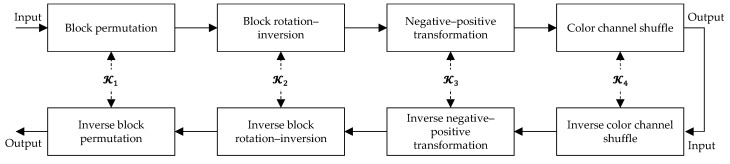
An illustration of block-based CPE encryption and decryption processes, where each $K_{i},\;i=1,\ldots ,\;4$ is a set of keys used in each step to process the color channels.

**Figure 4 sensors-23-04057-f004:**
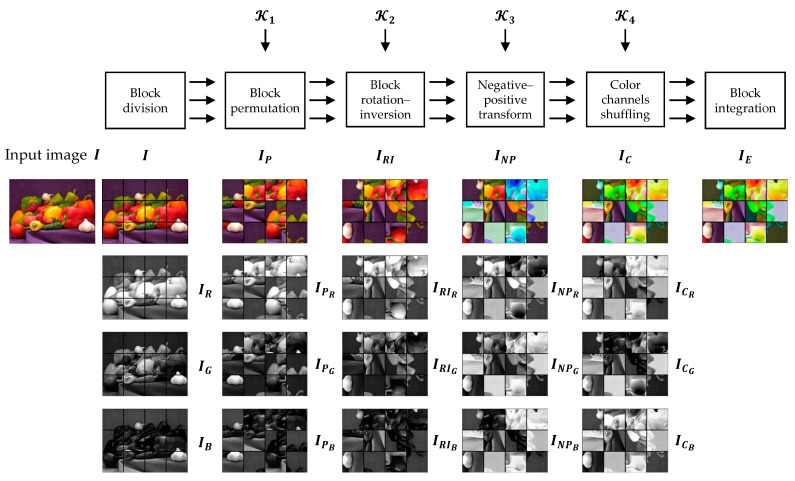
The encryption algorithm steps of a Color CPE scheme. For visual analysis, the effect of each transformation function on the image is shown across each color channel. The keys $K_{i},\;i=1,\ldots ,\;4$ is a set of keys used in each step to process the color channels. Because a common key is used to process each color channel, the blocks have the same appearance in each channel.

**Figure 5 sensors-23-04057-f005:**
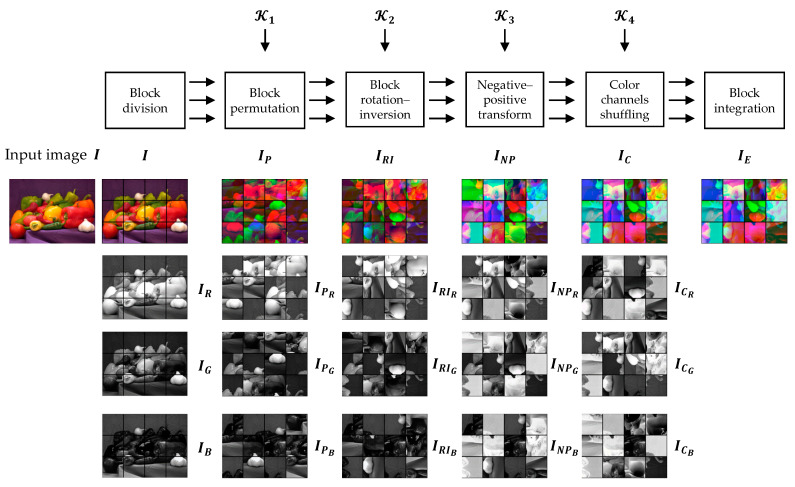
The encryption algorithm steps of an Extended CPE scheme. For visual analysis, the effect of each transformation function on the image is shown across each color channel. The keys $K_{i},\;i=1,\ldots ,\;4$ is a set of keys used in each step to process the color channels. Each color component is processed independently; therefore, the blocks have different appearances.

**Figure 6 sensors-23-04057-f006:**
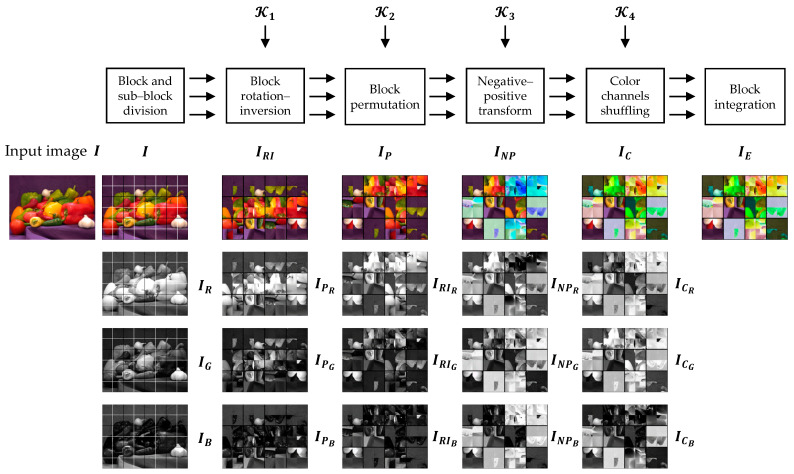
The encryption algorithm steps of an IIB–CPE scheme. The black line shows block division, whereas the white line shows sub-block division. For visual analysis, the effect of each transformation function on the image is shown across each color channel. The keys $K_{i},\;i=1,\ldots ,\;4$ is a set of keys used in each step to process the color channels. The local contents in each block are scrambled because of the sub-block processing.

**Figure 7 sensors-23-04057-f007:**
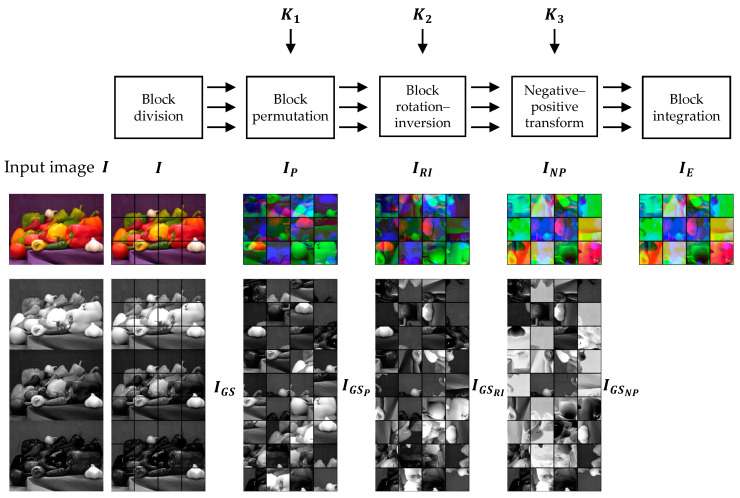
The encryption algorithm steps of a PGS–CPE scheme. The pseudo-grayscale representation was obtained by concatenating the RGB components along the vertical axis. For visual analysis, the effect of each transformation function on the image is shown. The keys $K_{i},\;i=1,\ldots ,\;3$ is used in each step to transform the blocks. The appearances of the blocks are similar to those from the methods that process the color component independently.

**Figure 8 sensors-23-04057-f008:**
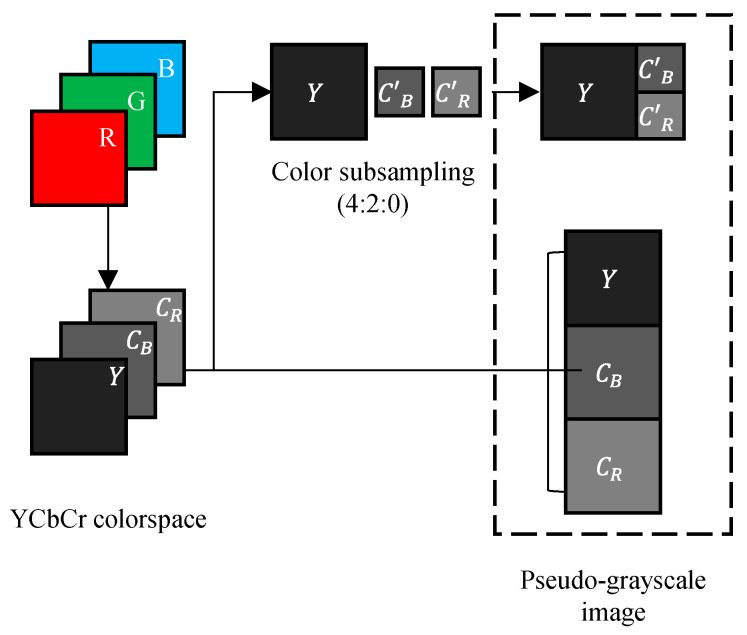
The pseudo-grayscale image representation generation for both chroma subsampling and without chroma subsampling.

**Figure 9 sensors-23-04057-f009:**
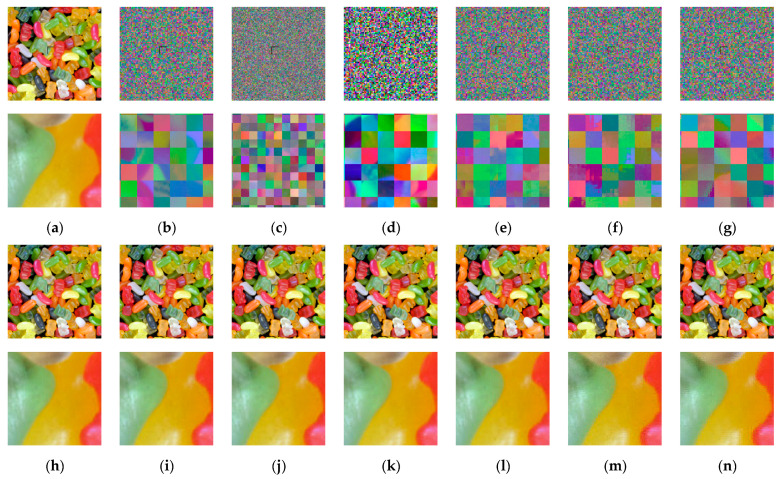
Visual analysis of the images recovered from the CPE processing. The JPEG algorithm was implemented without chroma subsampling. (**a**) The original image. (**b**–**g**) Cipher images obtained from the Color CPE, PGS–CPE, Extended CPE, IIB–CPE (8 × 8), IIB–CPE (4 × 4), and IIB–CPE (2 × 2) methods, respectively. The images recovered were compressed with the JPEG qf=71 in (**h**–**n**), qf=85 in (**o**–**u**), and qf=100 in (**v**–**ab**). For each image, the boxed region is zoomed in and shown below them.

**Figure 10 sensors-23-04057-f010:**
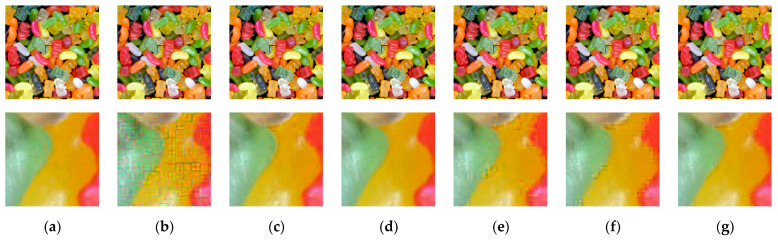
Visual analysis of the images recovered from the CPE processing. The JPEG algorithm with chroma subsampling (4:2:0) and the CPE were implemented on the image given in Figure 9. (**a**) The recovered image from the compression of plain image. (**b**–**g**) The recovered images from the compression of cipher images obtained from the Color CPE (8 × 8), Color CPE (16 × 16), PGS–CPE, IIB–CPE (8 × 8), IIB–CPE (4 × 4), and IIB–CPE (2 × 2) methods, respectively. The images recovered were compressed with the JPEG qf=71 in (**a**–**g**), qf=85 in (**h**–**n**), and qf=100 in (**o**–**u**). For each image, the boxed region is zoomed in and shown below them.

**Figure 11 sensors-23-04057-f011:**
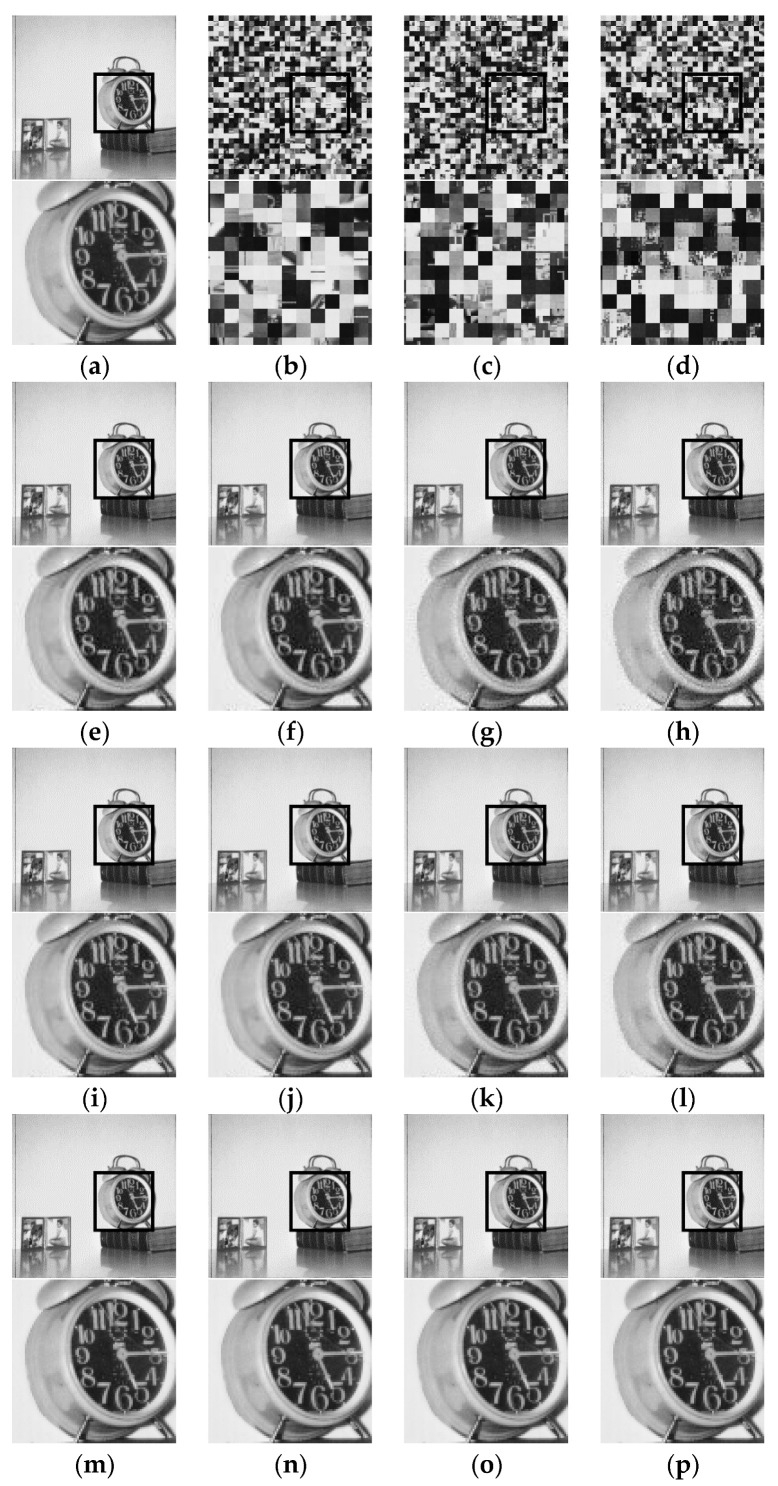
Visual analysis of the grayscale images recovered from the CPE processing. (**a**) The original image. (**b**–**d**) Cipher images obtained from the GS–CPE, GS–IIB–CPE (4 × 4), and GS–IIB–CPE (2 × 2) methods, respectively. The images recovered were compressed with the JPEG qf=71 in (**e**–**h**), qf=85 in (**i**–**l**), and qf=100 in (**m**–**p**). For each image, the boxed region is zoomed in and shown below them.

**Figure 12 sensors-23-04057-f012:**
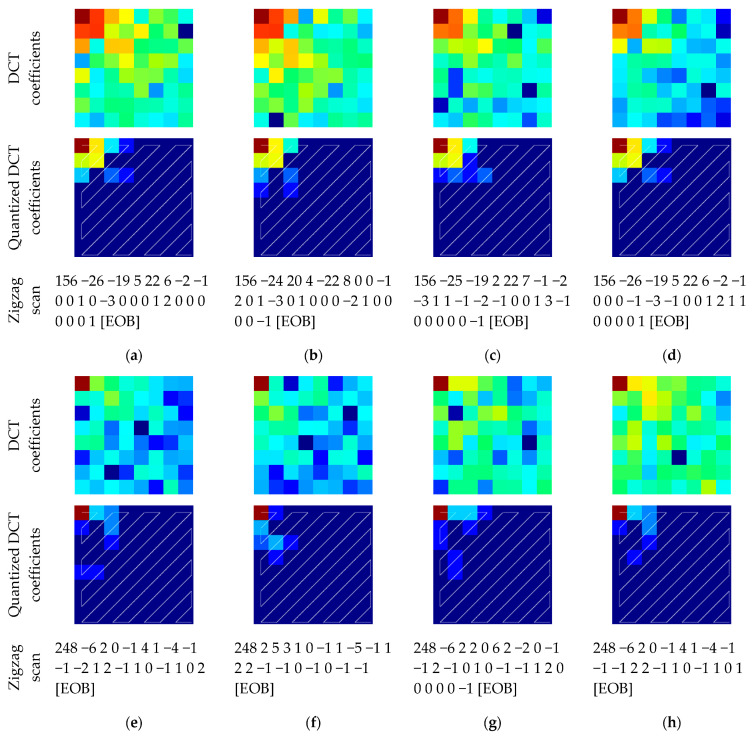
The CPE scheme compressibility analysis based on the DCT energy compaction. (**a**–**d**) The DCT of the block where correlation coefficients were $\sigma_{h}=0.95$ and $\sigma_{v}=0.96$. (**e**–**h**) The DCT of the block where correlation coefficients were $\sigma_{h}=0.49$ and $\sigma_{v}=0.52$. (**a**,**e**) The original block transformations. (**b**,**f**) The scrambled block transformations obtained by processing the entire blocks. (**c**,**g**) The scrambled block transformations obtained by the sub-block (4 × 4) processing. (**d**,**h**) The scrambled block transformations obtained by the sub-block (2 × 2) processing.

**Figure 13 sensors-23-04057-f013:**
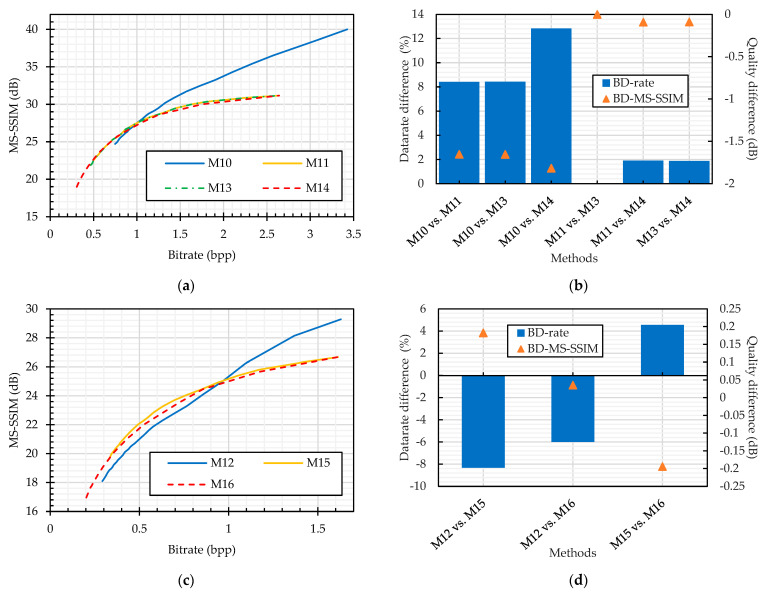
The JPEG compression analysis in color and grayscale mode. The JPEG compression is carried out without chroma subsampling in (**a**) and with chroma subsampling in (**c**); (**b**,**d**) are their corresponding BD-measures plots.

**Figure 14 sensors-23-04057-f014:**
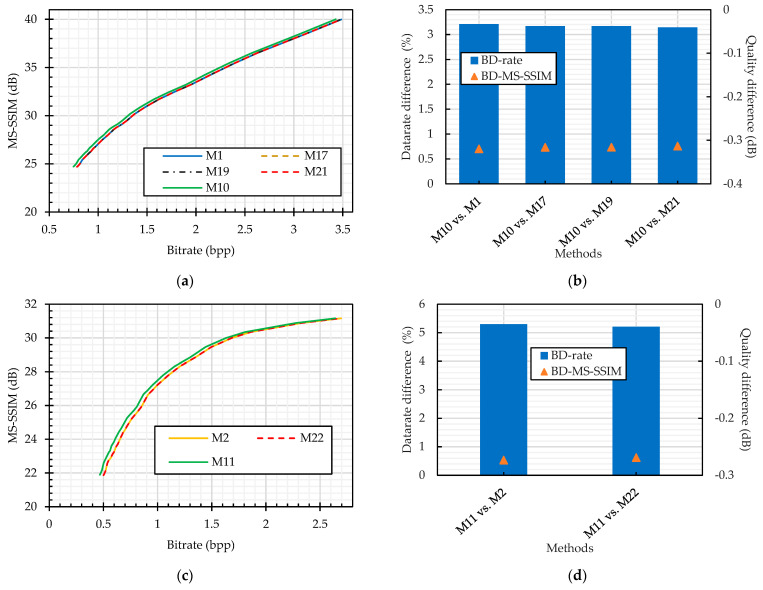
The JPEG compression analysis of plain and CPE images without chroma subsampling. The compression was carried out in RGB colorspace (**a**) and YCbCr colorspace (**c**); (**b**,**d**) are their corresponding BD-measures plots.

**Figure 15 sensors-23-04057-f015:**
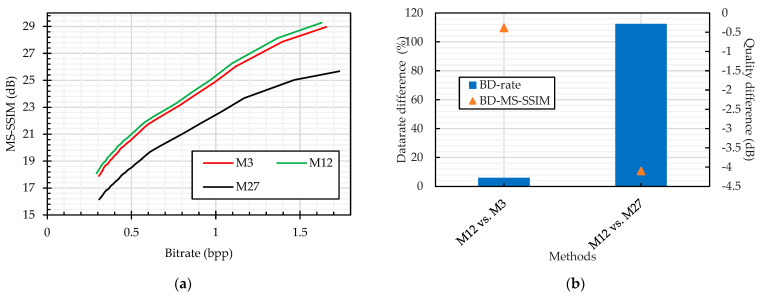
The JPEG compression analysis of plain and CPE images with chroma subsampling. (**a**) The RD curves and (**b**) the BD-measures plots.

**Figure 16 sensors-23-04057-f016:**
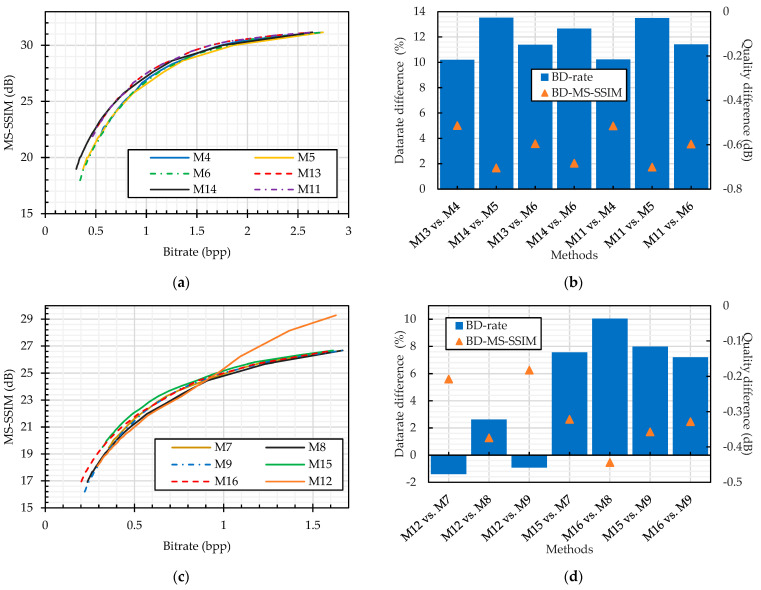
The JPEG compression analysis on plain and PGS–CPE images without and with chroma subsampling in (**a**) and (**c**), respectively; (**b**,**d**) are their corresponding BD-measures plots.

**Figure 17 sensors-23-04057-f017:**
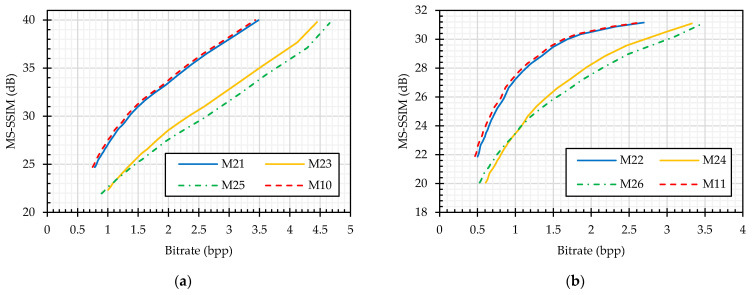
The sub-block size analysis in the IIB–CPE methods. In (**a**,**b**), the compression was carried out without chroma subsampling in the RGB and YCbCr colorspaces, respectively. (**c**) The compression was carried out with chroma subsampling. (**d**) The BD-measures plot.

**Figure 18 sensors-23-04057-f018:**
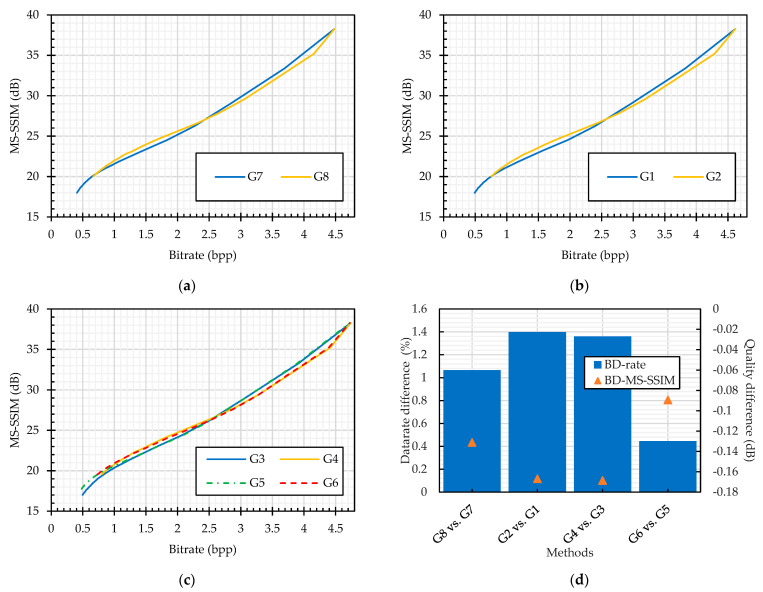
The JPEG compression analysis of plain and CPE grayscale images with respect to the quantization table choice. (**a**) Plain image compression (**b**) CPE schemes that perform entire block processing. (**c**) CPE schemes that perform sub-block processing. (**d**) The BD-measures plot.

**Figure 19 sensors-23-04057-f019:**
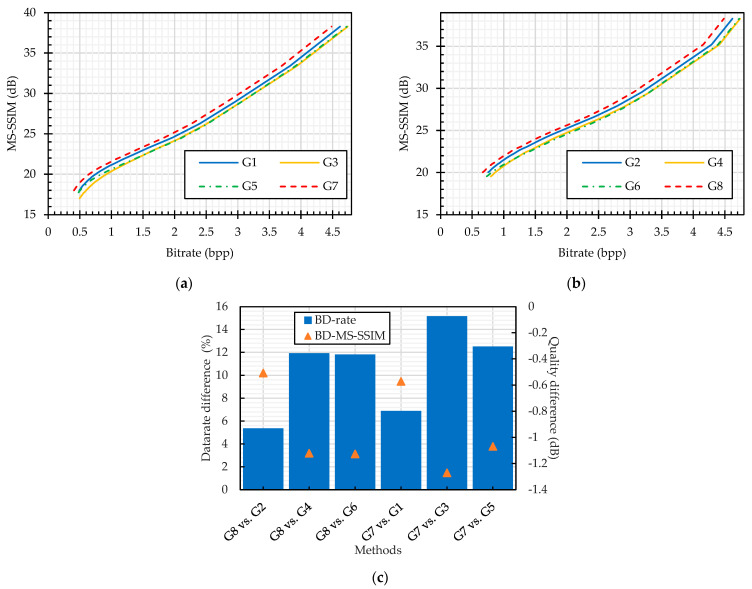
The JPEG compression analysis of plain and PE images. The compression was carried out using the chrominance table (**a**) and the luminance table (**b**). (**c**) is the BD–measures plot.

**Figure 20 sensors-23-04057-f020:**
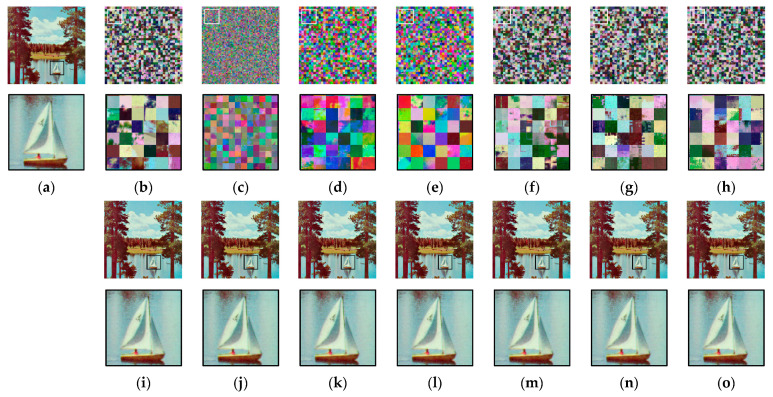
Visual analysis of the images recovered from the CPE processing. (**a**) The original image. (**b**–**h**) Cipher images obtained from the Color CPE, PGS–CPE, Extended CPE [47], Extended CPE [63], IIB–CPE (8 × 8), IIB–CPE (4 × 4), and IIB–CPE (2 × 2) methods, respectively. Their corresponding recovered images are shown in (**i**–**o**). For each image, the boxed region is zoomed in and shown below them. Note that the JPEG compression was not performed on the cipher images.

**Figure 21 sensors-23-04057-f021:**
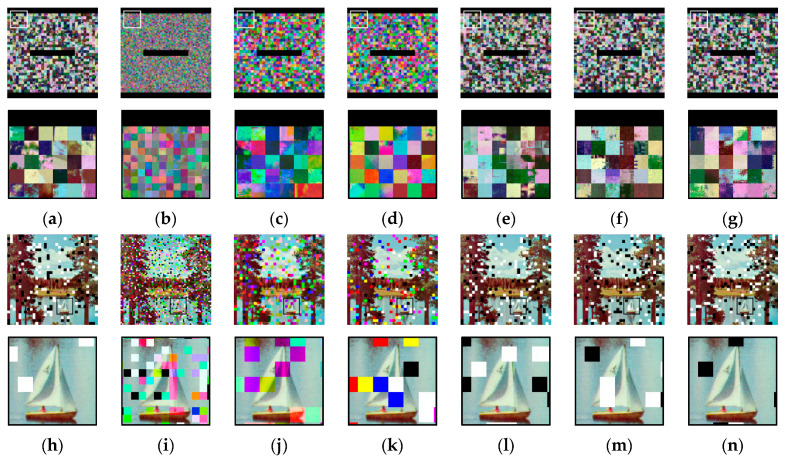
The CPE methods robustness against the data loss attack. (**a**–**g**) The cipher images given in Figure 20b–h with the data loss attack. Their corresponding recovered images are shown in (**h**–**n**). For better visual inspection, the boxed region in each image is enlarged and shown below them.

**Figure 22 sensors-23-04057-f022:**
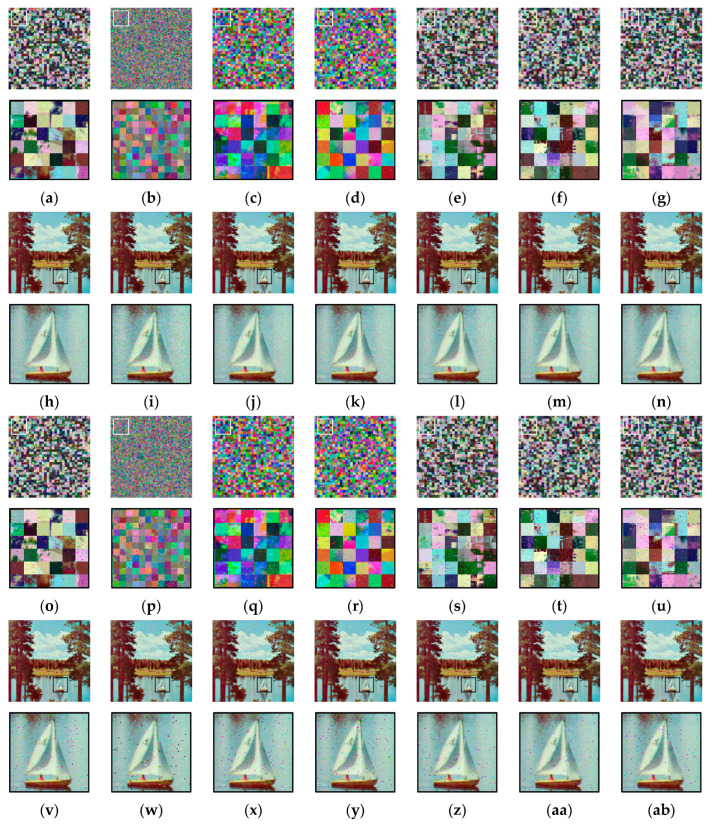
The CPE methods robustness against the noise attack. (**a**–**g**) and (**o**–**u**) are the cipher images given in Figure 20**b**–**h** with the noise attack by adding Gaussian and Salt–Pepper noises, respectively. The recovered images for (**a**–**g**) are shown in (**h**–**n**), and for (**o**–**u**) the recovered images are in (**v**–**ab**). For better visual inspection, the boxed region in each image is enlarged and shown below them.

**Table 1 sensors-23-04057-t001:** The IJG standard luminance quantization table.

16	11	10	16	24	40	51	61
12	12	14	19	26	58	60	55
14	13	16	24	40	57	69	56
14	17	22	29	51	87	80	62
18	22	37	56	68	109	103	77
24	35	55	64	81	104	113	92
49	64	78	87	103	121	120	101
72	92	95	98	112	100	103	99

**Table 2 sensors-23-04057-t002:** The IJG standard chrominance quantization table.

17	18	24	47	99	99	99	99
18	21	26	66	99	99	99	99
24	26	56	99	99	99	99	99
47	66	99	99	99	99	99	99
99	99	99	99	99	99	99	99
99	99	99	99	99	99	99	99
99	99	99	99	99	99	99	99
99	99	99	99	99	99	99	99

**Table 3 sensors-23-04057-t003:** A custom quantization table proposed in [48] for the JPEG algorithm without chroma subsampling.

17	26	32	39	46	54	67	90
26	35	42	50	56	65	80	105
34	43	51	58	65	75	91	118
42	53	60	68	76	86	103	131
50	62	69	77	86	98	116	145
61	73	81	90	99	112	133	164
76	90	99	108	118	133	157	192
98	116	126	136	147	165	193	233

**Table 4 sensors-23-04057-t004:** A custom quantization table proposed in [48] for the JPEG algorithm with chroma subsampling.

17	26	32	40	47	56	70	92
26	36	43	52	59	69	84	110
34	44	52	61	70	80	98	125
43	54	63	72	82	95	113	142
52	65	74	84	95	109	129	159
63	78	88	99	110	126	149	182
79	97	109	119	132	150	176	213
102	124	137	149	164	183	213	254

**Table 5 sensors-23-04057-t005:** CPE scheme implementation settings for color image encryption and compression.

Methods	Pseudonym	Input Image		Color Subsample	Quantization Table
		Colorspace	Image Type		
Color CPE	M1	RGB	Color	No	IJG Tables
	M2	YCbCr	Color	No	IJG Tables
	M3	RGB	Color	Yes	IJG Tables
PGS–CPE	M4	YCbCr	Pseudo-grayscale	No	IJG Luminance
	M5	YCbCr	Pseudo-grayscale	No	IJG Chrominance
	M6	YCbCr	Pseudo-grayscale	No	Custom [48]
	M7	YCbCr	Pseudo-grayscale	Yes	IJG Luminance
	M8	YCbCr	Pseudo-grayscale	Yes	IJG Chrominance
	M9	YCbCr	Pseudo-grayscale	Yes	Custom [48]
Plain Images	M10	RGB	Color	No	IJG Tables
	M11	YCbCr	Color	No	IJG Tables
	M12	RGB	Color	Yes	IJG Tables
	M13	YCbCr	Pseudo-grayscale	No	IJG Luminance
	M14	YCbCr	Pseudo-grayscale	No	IJG Chrominance
	M15	YCbCr	Pseudo-grayscale	Yes	IJG Luminance
	M16	YCbCr	Pseudo-grayscale	Yes	IJG Chrominance
Extended CPE [47]	M17	RGB	Color	No	IJG Tables
	M18	YCbCr	Color	No	IJG Tables
Extended CPE [63]	M19	RGB	Color	No	IJG Tables
	M20	YCbCr	Color	No	IJG Tables
IIB–CPE (8 × 8)	M21	RGB	Color	No	IJG Tables
	M22	YCbCr	Color	No	IJG Tables
IIB–CPE (4 × 4)	M23	RGB	Color	No	IJG Tables
	M24	YCbCr	Color	No	IJG Tables
IIB–CPE (2 × 2)	M25	RGB	Color	No	IJG Tables
	M26	YCbCr	Color	No	IJG Tables
IIB–CPE (8 × 8)	M27	RGB	Color	Yes	IJG Tables
IIB–CPE (4 × 4)	M28	RGB	Color	Yes	IJG Tables
IIB–CPE (2 × 2)	M29	RGB	Color	Yes	IJG Tables

**Table 6 sensors-23-04057-t006:** CPE scheme implementation settings for grayscale image encryption and compression.

Methods	Pseudonym	Quantization Table
GS–CPE	G1	IJG Luminance
	G2	IJG Chrominance
GS–IIB–CPE (4 × 4)	G3	IJG Luminance
	G4	IJG Chrominance
GS–IIB–CPE (2 × 2)	G5	IJG Luminance
	G6	IJG Chrominance
Plain images	G7	IJG Luminance
	G8	IJG Chrominance

**Table 7 sensors-23-04057-t007:** The encryption analysis of the CPE schemes under different statistical tests.

Methods	Correlation Coefficient						Entropy	Histogram Variance
	Image Level			Block Level				
	Diagonal	Horizontal	Vertical	Diagonal	Horizontal	Vertical		
Plain	0.87	0.91	0.9	0.42	0.55	0.51	6.51	237.92
Color CPE	0.84	0.91	0.91	0.01	0	0	7.42	40.58
PGS–CPE	0.73	0.85	0.85	–0.01	0	0	6.83	112.01
Extended CPE [47]	0.84	0.91	0.91	0	0	0	7.42	40.59
Extended CPE [63]	0.84	0.91	0.91	0	0	0	7.42	40.59
IIB–CPE (8 × 8)	0.83	0.9	0.9	0	0.01	0	7.42	40.6
IIB–CPE (4 × 4)	0.82	0.89	0.89	0	0	0	7.42	40.6
IIB–CPE (2 × 2)	0.83	0.9	0.89	0.01	0.01	0	7.42	40.57

**Table 8 sensors-23-04057-t008:** The CPE schemes robustness analysis against the JPS attack.

Methods	*D_c_*	*N_c_*	*L_c_*
Color CPE	0.005	0.111	0.120
PGS–CPE	0.001	0.001	0.002
Extended CPE	0.004	0.006	0.008
IIB–CPE (8 × 8)	0.01	0.08	0.02
IIB–CPE (4 × 4)	0.01	0.05	0.02
IIB–CPE (2 × 2)	0.01	0.06	0.02

**Table 9 sensors-23-04057-t009:** Quality of the recovered images under the different types of loss attacks shown in Figure 21 and Figure 22.

Methods	Data Loss	Gaussian Noise	Salt–Pepper Noise
Color CPE	0.54	0.95	0.91
PGS–CPE	0.24	0.88	0.86
Extended CPE [47]	0.54	0.95	0.91
Extended CPE [63]	0.55	0.95	0.91
IIB–CPE (8 × 8)	0.55	0.95	0.91
IIB–CPE (4 × 4)	0.54	0.95	0.91
IIB–CPE (2 × 2)	0.54	0.95	0.91

## Data Availability

All the datasets used in this study are publicly available. The Tecnick dataset used for color image compression and encryption is accessible at: https://testimages.org/ (accessed on 16 December 2021). The Shenzhen dataset used for grayscale image compression analysis is accessible at: https://ceb.nlm.nih.gov/repositories/tuberculosis-chest-X-ray-image-data-sets/31 (accessed on 13 March 2022). The USC-SIPI Miscellaneous dataset is accessible at: https://sipi.usc.edu/database/database.php?volume=misc (accessed on 4 July 2022).

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
