# Peer review of "Comprehensive Analysis of Compressible Perceptual Encryption Methods—Compression and Encryption Perspectives"

_sensors, 2023, doi:10.3390/s23084057_

Round 1

Reviewer 1 Report

The idea discussed in this paper is relatively new, however, I have a few concerns. 

1. Please discuss numeric values in the abstract. 

2. The main issue in this work is, why encryption then compression? Why not compression before encrypting an image. Please provide 

3. In Fig. 1, security and usability are vague and unclear.  

4. What is negative/positive transform? 

5. What is the negative correlation in the case of digital images?

6. Entropy should be 8 ideally. But in your case, it is around 7.5?

7. It would be nice to add histogram plots. 

8. If possible, please provide GPU resource details on acknowledgement. 

Reviewer 2 Report

In this review, a comprehensive analysis of compressible perceptual encryption methods, with the aim to be JPEG compatible. In their proposal, they present a comprehensive analysis of the JPEG compatible with compressible perceptual encryption methods in terms of their encryption and compression efficiencies. In my personal point of view, this proposal is interesting, the structure of this paper is organized well and is easily readable. However, there are some simple issues that need to be explained. I have the following observations: 

1-    In page 4, I suggest writing just Figure 1, i. e., without the last period.

2-    Since the authors are considering compression issues, there exist different transforms that play a key role, such as the wavelet transform, but it is not mentioned in any way. In fact, this transform has shown to be more efficient that many compressions techniques. Why is this transform not considered?

3-    Some captions of figures need to be more explained.

4-    Even though the proposal has been evaluated with a comprehensive set of metrics, it will be interesting if the authors consider the 2D-DFA as an objective metric since some metrics sometimes present some drawbacks to having an objective visual security. 

I recommend this paper to be accepted after to attend the observations.

Round 2

Reviewer 1 Report

I am satisfied with all answers.